# Biological effects of the loss of homochirality in a multicellular organism

Agnes Banreti [1] ✉, Shayon Bhattacharya [2,7], Frank Wien[3,7], Koichi Matsuo [4], Matthieu Réfrégiers [5], Cornelia Meinert [6], Uwe Meierhenrich [6], Bruno Hudry [1,7], Damien Thompson [2,7] & Stéphane Noselli[1]

Homochirality is a fundamental feature of all known forms of life, maintaining biomolecules (amino-acids, proteins, sugars, nucleic acids) in one specific chiral form. While this condition is central to biology, the mechanisms by which the adverse accumulation of non-L-α-amino-acids in proteins lead to pathophysiological consequences remain poorly understood. To address how heterochirality build-up impacts organism's health, we use chiral-selective in vivo assays to detect protein-bound non-L-α-amino acids (focusing on aspartate) and assess their functional significance in *Drosophila*. We find that altering the in vivo chiral balance creates a 'heterochirality syndrome' with impaired caspase activity, increased tumour formation, and premature death. Our work shows that preservation of homochirality is a key component of protein function that is essential to maintain homeostasis across the cell, tissue and organ level.

Homochirality is an overarching design principle in all living organisms[1–5]. RNA and DNA are built from D-sugars and protein biopolymers from L-α-amino acids (α-linked L-amino acids, hereinafter referred to as L-AAs), with any departure from this primary condition being detrimental to life. For example, random mixing of L-AAs and D-AAs within proteins would generate an almost infinite, uncontrolled combinatorial diversity of the proteome that would produce harmful phenotypes[6–8]. Therefore, various chiral checkpoints ensure that exclusively L-AAs are incorporated during translation[8–11]. These include aminoacyl-tRNA synthetases, elongation factor Tu and ribosome[12–18], and chiral proofreading by D-aminoacyl-tRNA deacylase[19–25]. D-AAs were long believed to be non-proteinogenic but have been found in the peptides of microorganisms[26]. Loss of proteome enantiopurity has also been detected in higher organisms including human[5,7,27], with recent studies pointing to various factors such as UV irradiation[28] or oxidative stress[29–31] as triggers of post-translational diastereoisomerisation of AA residues. In particular, D-, iso-L-, and iso-D-aspartyl residues (where iso-aspartyl is a β-linked amino acid that results from the

degradation of aspartic acid) have been detected in diverse proteins from various tissues of elderly individuals[29,32–34], including amyloid-β in the brain, crystallin in the lens, and elastin in the skin[30,35–39], which indicates a link between aspartyl heterochirality of proteins and age-associated disorders such as neurodegeneration, chronic kidney disease and cataracts[36,39–41]. These observations indicate that active control of homochirality is a key requirement for homoeostasis of living organisms. Protein-L-isoaspartate (D-aspartate) O-methyltransferase (Pimt) is an enzyme essential for protein repair by recognising and converting abnormal D-aspartyl[42,43] and iso-L-aspartyl[44,45] residues back to their native L-aspartyl form[12,46–50]. Pimt activity has been detected in a wide variety of organisms including bacteria[51–55], *C. elegans*[56], and various human tissues[43,50,57,58]. Futhermore, enzymatic activity of Pimt isolated from bacteria[59], human[60], *Drosophila*[61], and *Arabidopsis*[62] has been assayed. Accumulation of Pimt's protein substrates has been detected in *Pimt* deletion mutant *C. elegans*[63,64], null mutant mice[41,65], and plants[66,67]. Pimt has been linked to stress tolerance in plants[67], and to increased survival under abiotic stress conditions in *C. elegans* and

[1]Université Côte d'Azur, CNRS, Inserm, Institut de Biologie Valrose, 06108 Nice, France. [2]Department of Physics, Bernal Institute, University of Limerick, V94 T9PX Limerick, Ireland. [3]DISCO Beamline, Synchrotron SOLEIL, 91192 Gif-sur-Yvette, France. [4]HiSOR Hiroshima Synchrotron Radiation Center, Hiroshima University, Hiroshima, Japan. [5]Centre de Biophysique Moléculaire, CNRS; UPR4301, 45071 Orléans, France. [6]Université Côte d'Azur, Institut de Chimie de Nice, CNRS; UMR 7272, 06108 Nice, France. [7]These authors contributed equally: Shayon Bhattacharya, Frank Wien, Bruno Hudry, Damien Thompson. ✉e-mail: Agnes.Banreti@univ-cotedazur.fr

*Drosophila*[63,68]. Importantly, *Pimt* knock-out mice showed significant growth retardation succumbing to fatal seizures at an average of 42 days after birth[41], and increased proliferation and granule cell number in the dentate gyrus[69]. Pimt expression and enzyme activity were significantly decreased in human astrocytic tumours[70] and promoted epithelial mesenchymal transition in lung adenocarcinoma cell lines[71,72], indicating that impaired Pimt activity has several pathophysiological consequences.

Despite extensive studies focusing on the molecular mechanisms of Pimt-mediated protein repair, no direct biological link to the partial loss of homochirality has been shown due to the lack of tools to detect non-L-α-aspartyl residues in vivo and in situ. Furthermore, a direct link between the partial loss of homochirality and protein dysfunction has not been shown, and hence the underlying molecular and cellular mechanisms connecting heterochirality to pathophysiological sequelae remains unknown.

## Results

### Chiral-selective antibodies to detect heterochirality in vivo

To dynamically manipulate, trace, and characterise the pattern of proteome chirality, we first established a chiral-deficient in vivo model, in which animals accumulate non-L-AAs, and hence become heterochiral (Fig. 1a). This was achieved by generating a *Drosophila Protein-L-iso-aspartate (D-aspartate) O-methyltransferase* (*Pimt*) null mutant (*Pimt*^null1 or *Pimt*^n1), in which the entire gene locus was deleted using CRISPR-Cas9-directed genome editing (Fig. 1a and Supplementary Fig. 1a). Generating a knock-in *Pimt-Gal4* into the genomic *Pimt* locus (Supplementary Fig. 1a) allowed us to show that *Pimt* is ubiquitously expressed in tissues throughout the fly life cycle (Supplementary Fig. 1b).

To assess the degree of heterochirality produced in *Pimt*^n1 mutants, we generated chiral-specific antibodies that detect oligopeptide sequences with mixed chirality. Among all chiral AA protein residues, L-aspartyl (Asp, D) has the fastest isomerisation and epimerisation rate especially when present upstream of a glycine (Gly, G) residue[73–79].

Thus, isomerised and epimerised aspartyl residues are the most abundant non-L-amino acids in proteins.

Caspases (cysteine-dependent aspartate-specific proteases) and Granzymes are among the few proteases known to specifically hydrolyse peptide bonds following aspartyl residues. Caspases play a key role in mediating apoptotic cell death and show high substrate selectivity. They can selectively hydrolyse the only few among many thousands of proteins through recognition and hydrolysis of only one (rarely two) consensus sequence present in their target proteins[80–82].

With a canonical DEV**D**G motif, i.e., {L-Asp}(P4)-{L-Glu}(P3)-{L-Val}(P2)-{L-Asp}(P1)-{L-Gly}(P1′), caspase consensus cleavage sites provide natural "hot-spots" for chiral stereoinversions of L-aspartyl residues. We thus raised antibodies against homochiral and heterochiral versions of the caspase cleavage motif in which the aspartate at the (P1) position was converted into a D-form (DEV***d***G, ***d*** = D-Asp), an iso-L-form (DEV***βD***G, ***βD*** = ***isoD*** = iso-L-Asp), or an iso-D-form (DEV***βd***G, ***βd*** = ***isod*** = iso-D-Asp) (Fig. 1b).

Antibodies were tested by immunohistochemistry on *Pimt* mutant tissues (Fig. 1b, c; Supplementary Fig. 2a, b). As shown in Fig. 1b, c, the adult gut of the heterochiral *Pimt*^n1 animals showed strong immunoreactivity with the anti-DEV***d***G and anti-DEV***βD***G, in contrast to the homochiral controls. This was further confirmed by western blot analysis (Supplementary Fig. 2c, d). While a weak signal was detected with the DEV***βd***G-specific antibody, the difference between *Pimt* mutant and control is insignificant, suggesting that DEV***βd***G residues are not targeted by Pimt. No reactivity was observed with the anti-DEV**D**G antibody in either the homochiral or heterochiral animals, suggesting that the homochiral epitope is not exposed to the surface in target proteins, hence diastereoisomerisation might induce conformational changes to form diverse heterochiral neoepitopes (see

below). This is consistent with previous studies showing that certain caspase target sequences are only detectable immunologically following caspase-mediated proteolytic cleavage, which expose neoepitopes of the target protein surface (e.g., TubulinΔCsp6, cleaved Caspase-3 or Fractin antibodies[83,84]).

### Caspase cleavage sites are targets of Pimt-mediated repair

Next, we tested if Pimt has the capacity to recognise, methylate and repair isomerised DEVDG substrates. We performed a Pimt methyltransferase enzyme kinetics assay (Fig. 1d), using synthetic DEVDG-based homochiral and heterochiral oligopeptides from PARP1, a naturally occurring human Caspase-3 target protein (homochiral oligopeptide Ac-KRKGDEVDGVDEVAK-amide and heterochiral oligopeptide Ac-KRKGDEV***βD***GVDEVAK-amide). Results show an active Pimt enzymatic activity when using the heterochiral substrate, while no enzymatic activity was detected with the homochiral control peptide (Fig. 1d). These data demonstrate that Pimt specifically binds to isomerised DEVDG substrates and further support that such substrates accumulate in Pimt mutant animals.

### Heterochirality induces the formation of neo-epitopes

The stereospecificity of the four chiral antibodies was further validated by chiral-specific ELISA (Fig. 2a). The results shown confirm that the four homochiral and heterochiral epitope specific antibodies have no cross-reactivity with one another. This also suggests that incorporation of a single non-L-α-amino acyl residue into caspase cleavage sites induces structural changes compared to the homochiral peptide epitope.

### Heterochirality induces structural changes

To confirm this, the chemical structures of the homochiral and heterochiral synthetic antigens were characterised using synchrotron radiation circular dichroism spectroscopy (SRCD), a method highly sensitive to chiral changes (Fig. 2b and Supplementary Fig. 3a–c). Analysis of the far-UV (175 to 260 nm) SRCD spectra showed a wavelength shift for Ac-[CGG]-DEV***d***G-amide at 197 and 213 nm compared to the homochiral control Ac-[CGG]-DEV**D**G-amide (Fig. 2b and Supplementary Fig. 3c).

Overall, these data indicate that it is possible to raise highly specific enantio-selective antibodies against all four protein-bound forms of aspartate, which provides a valuable tool to characterise the in vivo chiral status of cells in any system.

### Heterochirality affects caspase-mediated cleavage of targets

Having shown that DEVDG caspase cleavage sites are potential targets of chiral deviation, we explored how creation of a single chiral AA variant *via* the stereoinversion of a single amino acid residue into D-, iso-L-, or iso-D-form alters the biochemical properties, including stability and resistance to proteolysis by caspases, which, consequently, could impair programmed cell death. To this goal, we first developed a chiral-specific in vitro caspase cleavage assay (DEVDase; ChiCAS cleavage assay), which is based on the release by recombinant human Caspase-3 of a fluorescent moiety (AMC) from the hydrolysis of a synthetic fluorogenic pentapeptide probe, Ac-DEVD-AMC (acetyl-{L-Asp}(P4)-{L-Glu}(P3)-{L-Val}(P2)-{L-Asp}(P1)−7-amido-4-methylcoumarin) (Fig. 3a). Comparing homochiral and heterochiral fluorogenic caspase substrates, we show that in contrast to the positive control substrate (Ac-DEVD-AMC), Caspase-3 was unable to cleave probes containing D-, iso-L-, or iso-D-aspartyl-residues (Fig. 3b). As homochiral negative controls, we used non-cleavable Ac-DEVA-AMC and Ac-DEVG-AMC substrates. These results clearly indicate that caspases are unable to cleave heterochiral substrates in vitro.

The atomic-scale mechanism for this malfunction was revealed by extensive molecular dynamics (MD) computer simulations (Fig. 3c−e, Supplementary Figs. 6−8 and Supplementary Tables 1, 2). We show that

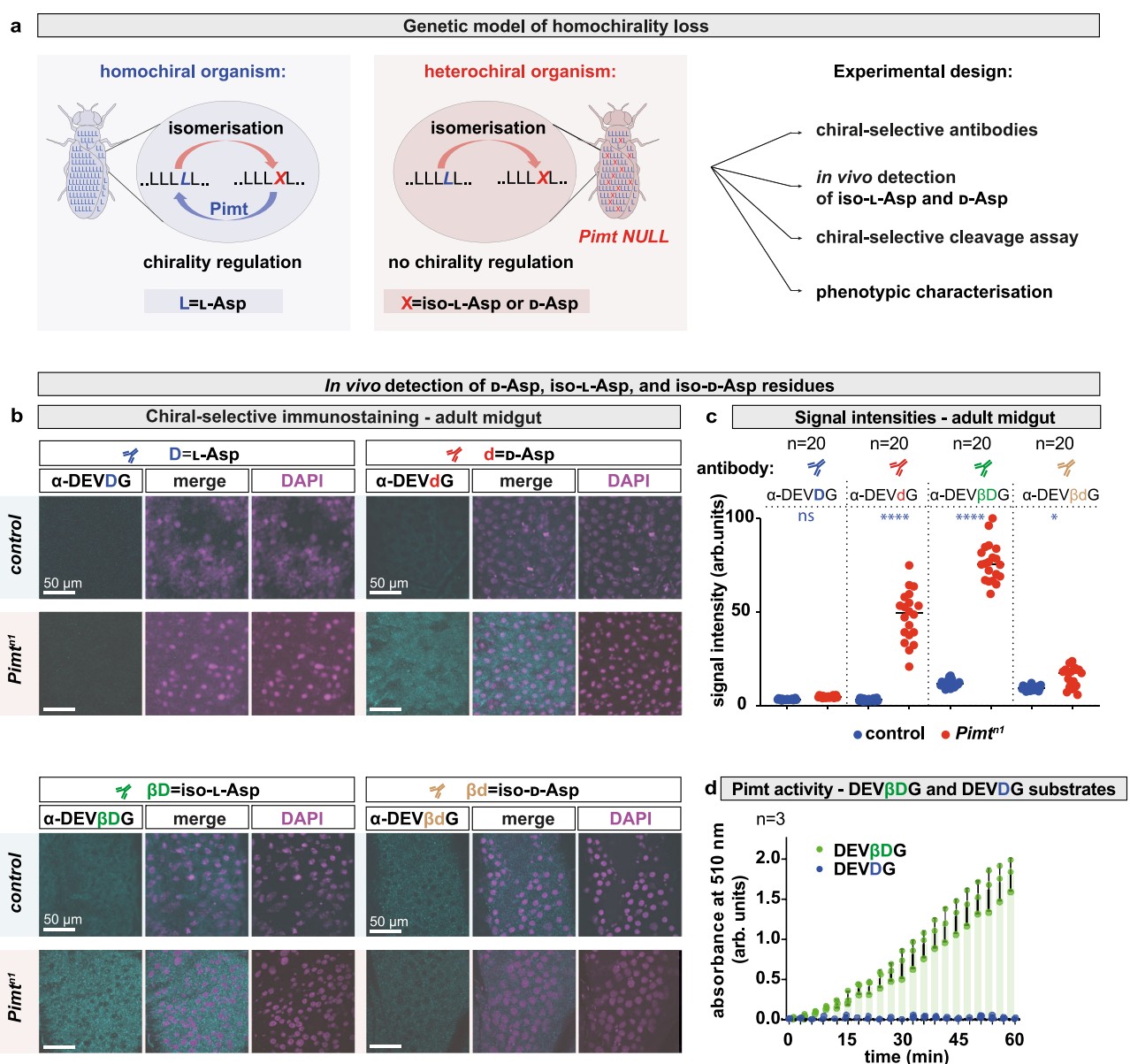

**Fig. 1 | Development of chiral-selective antibodies to detect heterochirality in vivo. a** Schematic representation of Pimt enzymatic activity and experimental design to assess the physiological effect of loss of protein homochirality. Animals lacking Pimt activity accumulate heterochiral proteins due to the lack of correction of post-translational chiral errors. For this and all subsequent figures, light blue shading highlights homochiral and light red shading highlights heterochiral animals. **b** Detection of hetero-neoepitopes in control and $Pimt^{n1}$ chiral-deficient 20 days-old animals. The anti-DEV*d*G, anti-DEV*βD*G, and anti-DEV*βd*G antibodies (where ***d*** corresponds to D-aspartate, ***βD*** to iso-L-aspartate and ***βd*** to iso-D-aspartate) were designed to recognise hetero-neoepitopes in the proteome containing D-, β-L-, and β-D-AA. As a control, anti-DEV*D*G was raised against a homochiral peptide. Dissected adult female guts were stained with anti-DEV*D*G (blue), anti-DEV*d*G (red), anti-DEV*βD*G (green), or anti-DEV*βd*G (yellow) antibodies. Guts of control animals did not show any immunoreactivity, while $Pimt^{n1}$ showed positive staining for the DEV*d*G-specific and DEV*βD*G-specific antibodies. **c** Quantification of normalised signal intensities of the four chiral-specific immunostainings shown in panel **b**. Experiments were repeated at least three independent times. *n* number of animals. Comparison of normalised signal intensities were done with One-way ANOVA test. *P* values for these and all subsequent statistical tests are defined as ****$p < 0.0001$; $0.0001 < $***$p < 0.001$; $0.001 < $**$p < 0.01$; $0.01 < $*$p < 0.05$; and *not significant* (ns) $0.05 < p$ and ns = 0.9783, *$p = 0.02$. **d** Pimt methyltransferase activity was assayed on the homochiral and heterochiral synthetic substrates Ac-KRKGDEVDGVDEVAK-amide and Ac-KRKGDEV*βD*GVDEVAK-amide (***βD*** corresponds to iso-L-aspartate), respectively. The sequence corresponds to the human PARP1 fragment outlined in the in vivo cleavage assay (Fig. 4a–d). The calculated methyltransferase activity at 37 °C on the linear portion of the curve is 0.06039 µmol/min/ml. Experiments were repeated three independent times. Values are presented as average ± standard deviation (S.D.). Source data are provided as a Source Data file for **c** and **d**.

the DEV*d*G chiral mutant can exit the caspase catalytic pocket after ~140 ns of dynamics (Fig. 3c), whereas DEV*D*G remains bound keeping the L-Asp4′−Gly5′ peptide bond in close proximity to the catalytic His121 and Cys163 residues. DEV*d*G rejects Caspase-3 due to electrostatic repulsion (see Supplementary Fig. 7a, Supplementary Fig. 8b) of the heterochiral substrate, precluding cleavage of the D-Asp4′-Gly5′ peptide bond. In accordance with experimental measurements, the model predicts that the subtle change in geometry produced by a

single L-AA → D-AA substitution dramatically alters molecular recognition of the peptide sequence in the caspase binding pocket (Fig. 3e), indicating that Caspase-3 is stereoselective and hence rejects heterochiral target proteins (Fig. 3c–e).

## Chiral shift induces resistance to caspase-mediated cleavage

Our results show that caspases malfunction when the consensus cleavage site of target proteins suffer a stereoinversion, which could

**Characterisation of chiral-selective antibodies, homo- and heterochiral peptide epitopes**

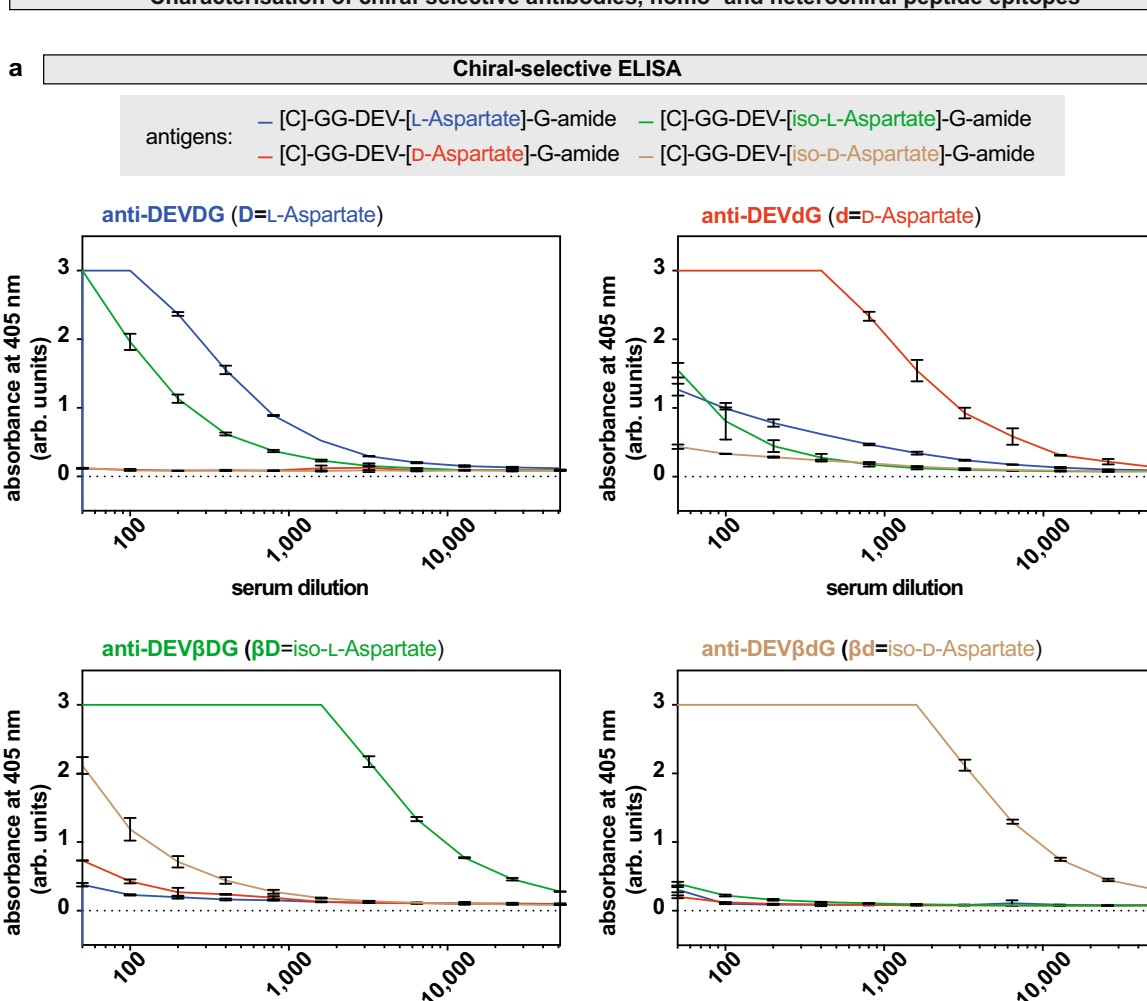

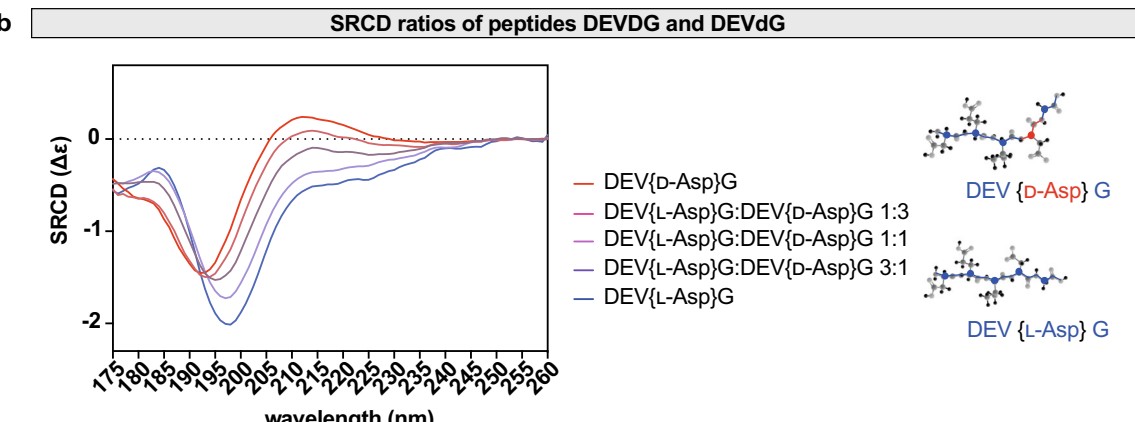

**Fig. 2 | Heterochirality induces structural changes and the formation of neo-epitopes. a** Cross-reactivity screening with ELISA. Anti-DEV*D*G (blue, *D* = L-Asp), anti-DEV*d*G (red, *d* = D-Asp), anti-DEV*βD*G (green, *βD* = iso-L-Asp), anti-DEV*βd*G (yellow, *βd* = iso-D-Asp) antibodies were raised against the corresponding homochiral and heterochiral oligopeptide epitopes. Antibody−antigen interactions were measured using a colorimeter and absorbance at 405 nm is shown as a function of serum dilution. Experiments were repeated three independent times. Values are presented as average ± standard deviation (S.D.). **b** Recorded far-UV synchrotron radiation circular dichroism (SRCD) spectra of synthetic oligopeptides: [C]-GG-DEV[L-Asp]{P1}G-amide L-enantiopure control (blue), heterochiral [C]-GG-DEV[D-Asp]{P1}G-amide peptide in which the P1 aspartate is in D-form (red) and mixtures of homochiral and heterochiral oligopeptides in the indicated ratios. Stick representation of the homochiral DEVDG and its stereoisomer heterochiral DEV*d*G pentapeptide, where the d in 4' position refers to D-Asp. Source data are provided as a Source Data file.

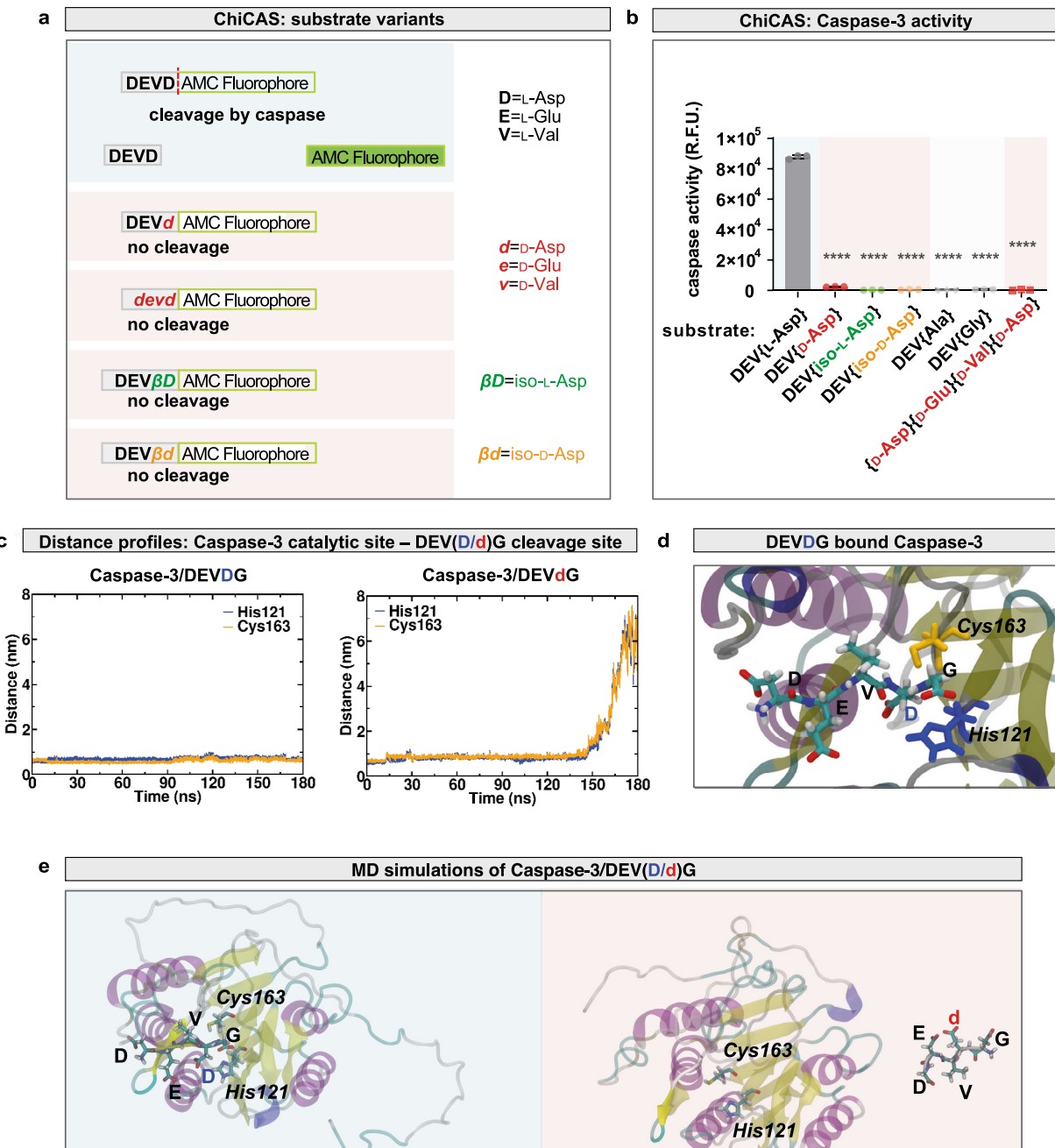

**Fig. 3 | Heterochirality affects caspase-mediated cleavage of target proteins.** Caspases are unable to cleave target sequences containing D-amino acid(s) as determined by chiral-specific DEVDase assays. **a** In vitro chiral-specific DEVDase assay (in vitro ChiCAS). Shown are the Ac-DEVD-AMC fluorogenic substrate consisting of all-L-AAs, substrates incorporating a single D-AA (Ac-DEV*d*-AMC), iso-L-AA (Ac-DEV*βD*-AMC), iso-D-AA (Ac-DEV*βd*-AMC) or consisting of all-D-AAs (Ac-*devd*-AMC). Letters in lower case denote D-AAs or iso-D-AAs. **b** Assay for substrate specificity of purified recombinant human Caspase-3 activity shows that the cleavage activity is stereoselective. Activity was followed in presence of the homochiral substrate and various heterochiral substrates: Ac-DEVD-AMC (homochiral control), Ac-DEV*d*-AMC, Ac-DEV*βD*-AMC, Ac-DEV*βd*-AMC, and Ac-*devd*-AMC (homochiral peptide with all-D-AAs). Negative controls Ac-DEV*A*-AMC and Ac-DEV*G*-AMC were also tested. Relative fluorescence units (R.F.U) were measured. Experiments were done in triplicates. Values are presented as average ± standard deviation (S.D.). *P*-values from two-sided Mann–Whitney U-test are ****$p < 0.0001$. Source data are provided as a Source Data file. **c** Distance profiles between Caspase-3 catalytic site and the peptide cleavage site computed from molecular dynamics (MD) simulations showing that the *d*-G peptide bond in DEV*d*G completely dissociates from the Cys163–His121 dyad making the heterochiral sequence inaccessible to nucleophilic attack from the protease enzyme. Source data files of distance profiles has been uploaded to 10.5281/zenodo.7199808. **d** The Caspase-3 binding pocket hosts the Cys163–His121 catalytic dyad that cleaves the D-G peptide bond of DEVDG. **e** Predictive models from MD simulations of DEV*D*G and DEV*d*G-bound Caspase-3 showing that the Cys163–His121 catalytic dyad remains inaccessible to DEV*d*G. Source data files of coordinates, topology and trajectory of the MD simulation are uploaded to https://doi.org/10.5281/zenodo.7199808.

Chiral-selective *in vivo* enzymatic cleavage assays

**a** ChiFCA: Isoform-specific *in vivo* cleavage assay

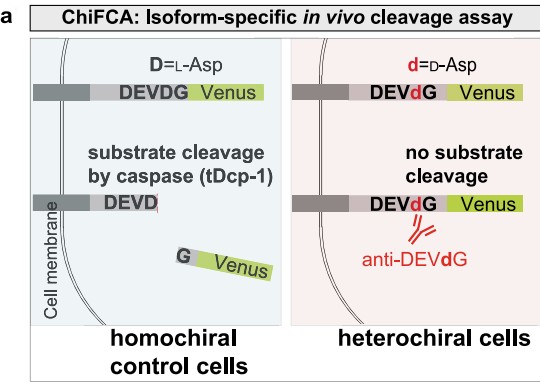

**b** *In vivo* chiral-selective cleavage assay - larval salivary gland

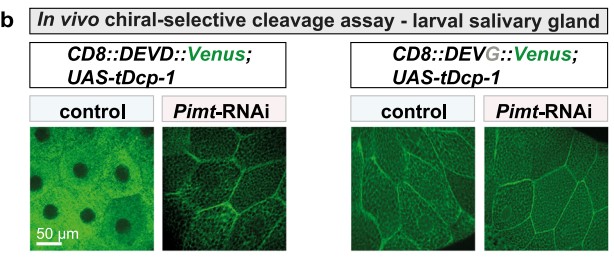

**c** Signal intensity ratios - larval salivary gland

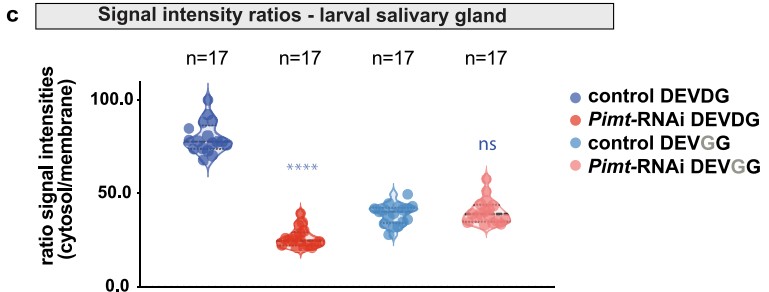

**d** Chiral-selective immunostaining - larval salivary gland

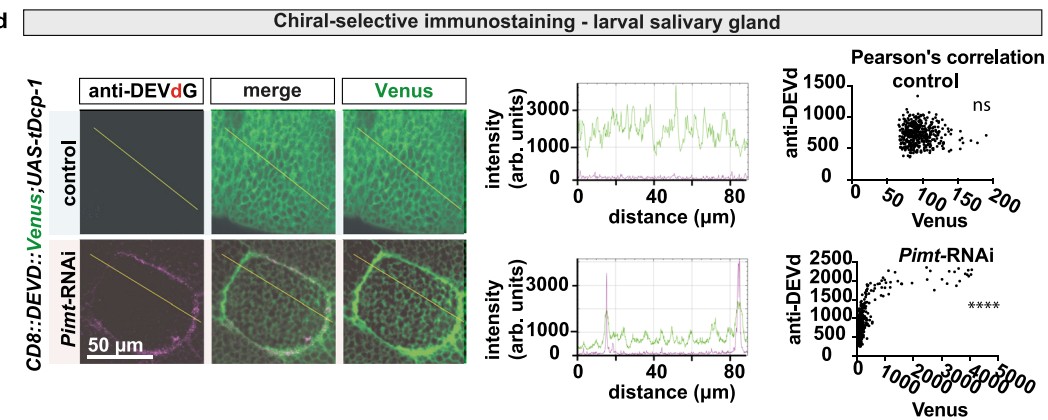

potentially affect many important cellular processes. To test this hypothesis, we developed a genetically encoded chiral-specific in vivo fluorometric caspase activity assay (in vivo ChiFCA assay, Fig. 4a) which confirmed that loss of homochirality results in the formation of substrates that are uncleavable by caspase proteins in vivo. The experiment uses an artificial effector caspase substrate (CD8::PARP1::Venus) in which a fragment of the human Poly (ADP-ribose) polymerase 1 protein (hPARP1, which contains the Caspase-3 consensus site DEVDG) is flanked by the transmembrane domain of CD8 protein at the N-terminus and the YFP Venus at the C-terminus (modified assay from ref. 85). Effector caspase activity was assayed by monitoring the translocation of membrane-bound fluorescent reporter protein into the cytoplasm. As control, an uncleavable construct with DEVGG sequence was also tested[85]. Apoptosis was induced in the

**Fig. 4 | Chiral shift induces resistance to caspase-mediated proteolytic cleavage in vivo. a** A schematic representation of the in vivo chiral-specific effector caspase activity (CHiFCA) assay (in vivo ChiFCA assay). The expressed transgenic construct consists of an artificial effector caspase substrate (including the Caspase-3 consensus site DEVDG) flanked by a transmembrane domain and the YFP Venus which is a specific reporter of effector caspase (DEVDase) activity during apoptosis in vivo (adapted and modified from ref. 85). Dcp-1 effector caspase (*Drosophila* functional homologue of Caspase-3) cleaves its substrates from the plasma membrane upon activation, with release of the YFP Venus-tagged C-terminal. **b** Confocal images of dissected salivary glands of homochiral (control) and heterochiral (*Pimt*-RNAi) larvae. Animals express *UAS-CD8::PARP1::Venus* and *UASp-tDcp-1* under the control of *nubbin-Gal4 (nub-Gal4)*. **c** Quantification of fluorescent signal intensities of Venus (green) at the cell membrane and in the cytosol, measured by ImageJ software. Note that all quantitative analyses were carried out on unadjusted raw images. Experiments were repeated at least three independent times, *n* number of cells for **b** and **c**. Values are presented as average ±standard deviation (S.D.). *P*-values from two-sided Mann–Whitney U-test are ****$p < 0.0001$ and ns = 0.8384. **d** Anti-DEV*d*G antibody was used to detect uncleaved DEV*d*G sequences in larval salivary gland cells expressing the PARP1 reporter. Fluorescent intensities of anti-DEV*d*G (red) and Venus (green) are measured by ImageJ software along the yellow lines. Experiments were repeated at least three independent times. Two-sided Pearson's correlation values were calculated, Pearson r for Venus versus anti-DEV*d*G signal intensities are −0.01988 (control), not significant and 0.6683 (*Pimt*-RNAi), ****$p$. Source data are provided as a Source Data file for **c** and **d**.

control and *Pimt*-RNAi animals, by ectopic expression of the activated form of the *Drosophila* effector caspase Dcp-1 (human Caspase-3 functional homologue). Our results show that in *Pimt*-RNAi cells, the CD8::PARP1::Venus substrate remained at the membrane, indicating that caspase activity was insufficient to cleave epimerised DEVDG substrates in *Pimt*-RNAi cells (Fig. 4b, c and Supplementary Fig. 4a, b). We could show co-localisation of our anti-DEV*d*G antibodies with the uncleaved, hence proteolytically resistant CD8::PARP1::Venus reporter, indicating that the accumulation of uncleaved caspase targets is linked to the presence of D-amino acids in the *Pimt*-RNAi cells (Fig. 4d and Supplementary Fig. 4c).

### Loss of homochirality reduces lifespan
Next, we investigated the biological consequences of loss of homochirality maintenance in cells. Importantly, *Pimt* knock-out flies showed premature death, dying 14 days earlier than control flies (Fig. 5a). Premature death was due solely to the lack of Pimt activity, as the phenotype could be fully rescued by a *Pimt* wild type (*Pimt^wt^*), but not a *Pimt* catalytic dead (*Pimt^S60Q^*) knock-in construct (Fig. 5a)[86], in which the evolutionarily conserved serine60 residue (Supplementary Fig. 5) was replaced by glutamine. Furthermore, we found that loss of Pimt activity led to the formation of protein aggregates and large melanotic tumours inside the body (Fig. 5b, c).

### Loss of homochirality enhances susceptibility to tumours
We next addressed how the loss of homochirality, which is shown above to induce caspase resistance, might in turn affect tumour formation, as one central hallmark of oncogenic cells is their ability to evade apoptotic cell death. We used a well-characterised, thermosensitive *Notch*-induced tumour model system in the progenitor cells (PCs) of the adult midgut[87]. At permissive temperature, *esg^TS^ > GFP, Notch*-RNAi flies (*esg-Gal4, tub-Gal80^TS^, UAS-Notch*-RNAi, where *esg* refers to *escargot* and *TS* refers to thermosensitive, respectively) develop tumours derived from the GFP-marked PCs in the posterior midgut (Fig. 6a, c, e). Importantly, when *Pimt* was simultaneously knocked down in PCs (*esg^TS^ > GFP, Notch*-RNAi, *Pimt*-RNAi), the total area of GFP-positive posterior midgut tumours was significantly increased relative to controls (Fig. 6b, d, e). In addition to increased tumour growth, we also observed the formation of ectopic tumours in the anterior region, a phenotype not observed in *esg^TS^ > GFP, Notch*-RNAi flies. These results thus indicate a strong tumour suppressor activity for Pimt. Remarkably, the downregulation of *Pimt* in *Notch*-RNAi PCs resulted in the accumulation of heterochiral proteins recognised by DEV*d*G and DEV*βD*G antibodies (Fig. 6f, g).

## Discussion
By genetically altering the conserved Pimt enzyme and developing specific genetic and cellular biology toolkits to detect and characterise proteome chiral changes, we reveal *Drosophila* as a unique animal model system to study the effect of heterochirality build-up in vivo. We based our experimental design on previous findings that isomerisation, and racemisation reactions do not arise randomly in protein sequences.

Instead, there are hot spots for generation of predominantly iso-L-Asp and D-Asp residues in sequences containing a small amino acid (glycine, serine or alanine residue) following aspartate[76,78,88].

We identified the conserved DEV*D*G caspase cleavage sites as important targets of chiral shift, leading to resistance to caspase activity, increased tumour susceptibility and a reduction in life expectancy supporting the view that the active maintenance of homochirality in *Drosophila* is critical for both apoptotic and non-apoptotic caspase activities and for possibly other yet unexplored, cellular processes. We showed that Pimt directly acts on DEV*D*G motifs for their chiral repair. Consequently, *Pimt KO* cells which have lost their ability to maintain homochirality, accumulate isomerised and epimerised aspartyl residues at consensus caspase cleavage sites. The corresponding neoepitopes can be immunologically detected. These posttranslational alterations at caspase cleavage sites provide a novel mechanism for the development of tumours as shown in this study, and potentially also a route to cancer cell resistance to apoptosis in higher organisms. In this respect, it is interesting to note that loss of caspase activity has been linked to cancer and ageing[89–91].

Based on the presence of a consensus DXXD/G (or S) sequence, hundreds of substrates for Caspase-3 have been identified, and possibly a thousand according to some estimations[82]. For example, PARP1 is cleaved and inactivated by caspases during almost all forms of apoptosis, and introduction of an uncleavable form of PARP1 can inhibit apoptotic progression[92,93]. Cleavage of PARP1 by Caspase-3 has been implicated in several neurological diseases -including Alzheimer's disease, multiple sclerosis, Parkinson's disease (Reviewed by Chaitanya et al.[94]). PARP1 is upregulated in many types of human cancer including ovarian, breast and pancreatic cancers, and several PARP inhibitors have been developed for clinical applications[95]. Hence, aspartate isomerisation and epimerisation which potentially render caspase-mediated processing of PARP1 (and likely other caspase targets) faulty, are expected to have massive physiological consequences on cell and tissue homoeostasis and might be implicated in many human diseases[96].

Overall, our results show that accumulation of non-L-α-AAs in proteins, promotes a progressive heterochirality syndrome, through a cascading effect across biological scales[97] spanning from loss of molecular homochirality to increased resistance to caspase activity in cells, increased tumour susceptibility in organs and, consequently, premature death of the chiral-deficient animal (Fig. 6h). We further suggest that heterochirality spreading in living organisms represents a novel causal factor that may be associated with a broad range of defective cellular processes, diseases and ageing. Developing methods for monitoring heterochirality in tissues, such as those described in this work, will provide new avenues for both basic and biomedical research and the study of cell signalling in altered chiral environments.

## Methods
### Fly strains
The following strains were used: *w^1118^* isogenic strain (from B. Hudry), *UAS-LacZ*-RNAi (2^nd^, from M. Miura[98]), *w*;P{UAS-Pimt.C}2-1* (2^nd^,

**Phenotypic characterisation of heterochiral animals**

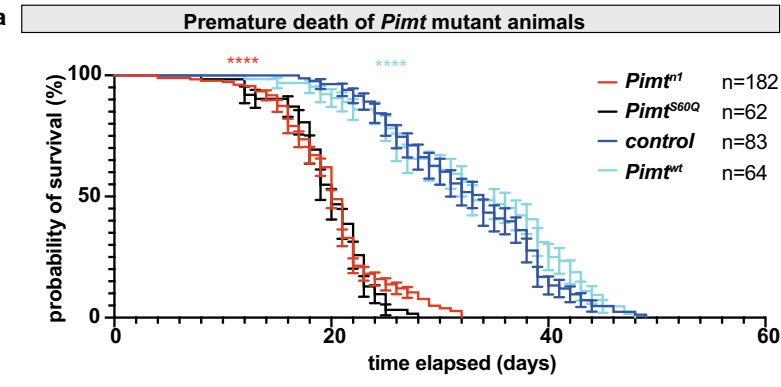

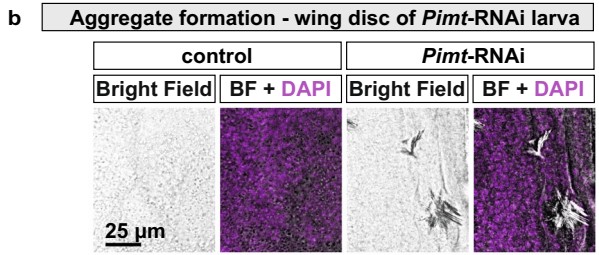

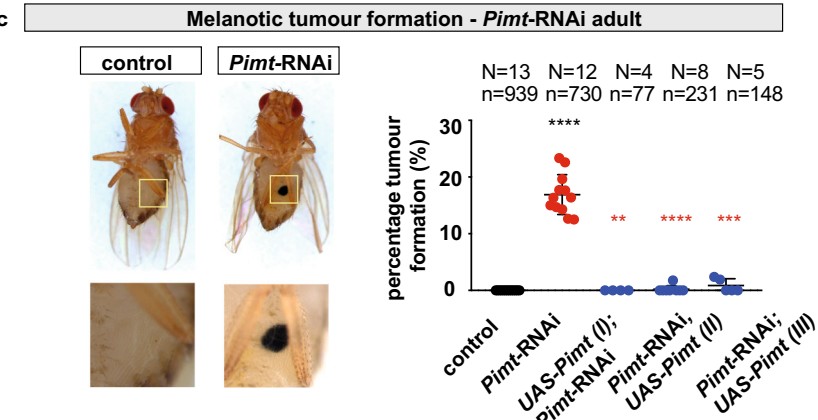

**Fig. 5 | Loss of homochirality reduces lifespan. a** Premature death of *Pimt^{n1}* knock-out null mutant animals is rescued by *Pimt* knock-in wild type (*Pimt^{wt}*) but not by *Pimt* knock-in S60Q (*catalytic dead, CD, Pimt^{S60Q}*) construct. Median survivals in days are: control: 34, *Pimt^{n1}*: 20, *Pimt^{wt}*: 33, *Pimt^{S60Q}*: 20. Experiments were repeated at least three independent times, *n* number of animals. Comparison of age-specific survival curves was done with Log-rank (Mantel−Cox) test. Chi-square values are: *Pimt^{n1}* to control (red): 140.6 (****p); *Pimt^{wt}* to *Pimt^{n1}* (light blue): 109 (****p); *Pimt^{wt}* to control: 0.9881 (p:0.3202); *Pimt^{S60Q}* to *Pimt^{n1}*: 0.9543 (p:0.3286). Values are presented as average ±standard deviation (S.D.). **b** Bright field (BF) and confocal

images represent protein aggregates in tissues of dissected heterochiral (*Pimt*-RNAi, right) and homochiral control animals (left). Experiments were repeated three independent times. **c** Control animals (left) and animals in which the chirality-regulating enzyme-encoding *Pimt* gene is downregulated, develop large melanotic masses (*Pimt*-RNAi, right). Percentage of animals developing melanotic tumour masses. *N* number of independent experiments, *n* number of animals. Values are presented as average ± standard deviation (S.D.). *P*-values from two-sided Mann−Whitney U-test are ****p < 0.0001, ***p = 0.0003 and **p = 0.0011. Source data are provided as a Source Data file for **a** and **c**.

Bloomington Drosophila Stock Centre, BDSC: 27393), *P{UAS-Pimt.C}X-2,w\** (1st, BDSC: 27394), *w\*;P{UAS-Pimt.C}3-2* (3rd, BDSC: 27395), *w\*;P{UAS-Pimt.IR}3−4* (3rd, BDSC: 27396), *P{UAS-Pimt.IR}X-5* (1st, BDSC: 27397), *Pimt^{null1}* (3rd, this study, #1-#8, 8 lines in total), *Pimt KI wt: Pimt^{wt}* (3rd, this study), *Pimt KI S60Q: Pimt^{S60Q}* (3rd, this study), *UAS-Pimt^{wt}-HA* (3rd, this study), *Pimt-Gal4* (3rd, this study), *Cre^{mini w+}* (3rd, BDSC: 1501), *ActGFP, Ser ^{mini w+}* (3rd, BDSC: 4534), *w^{1118};UAS-Stinger^{NLS}* (2nd, BDSC: 84277), *nub-Gal4* (2nd, BDSC: 25754), *Act5C-Gal4* (2nd, BDSC: 4414), *UAS-Pimt*-RNAi (2nd, Vienna Drosophila Resource Centre, VDRC: GD19123),

*UAS-Pimt*-RNAi (2nd, VDRC: GD19121), *UAS-Notch*-RNAi (3rd, VDRC: {GD14477}v27229), *esg-Gal4^{NP7397}, UAS-GFP, Tub-Gal80^{TS}* (2nd, from J. de Navascués), *UAS-CD8::PARP1 (DEVD)* and *UAS-CD8::PARP1 (DEVG)* (3rd, from Eli Arama[85]); *UASp-tDcp-1 (deltaN-Dcp-1)* (3rd, from Kim McCall[99]).

### Generation of *Pimt^{null1}* mutant

*Pimt^{null1}* mutant 5′ and 3′ homology arms (HAs) were cloned into pHD-DsRed-attP vector by gene synthesis (Genscript) using EcoRI and XhoI cloning sites, respectively. pHD-DsRed-attP is a vector for making

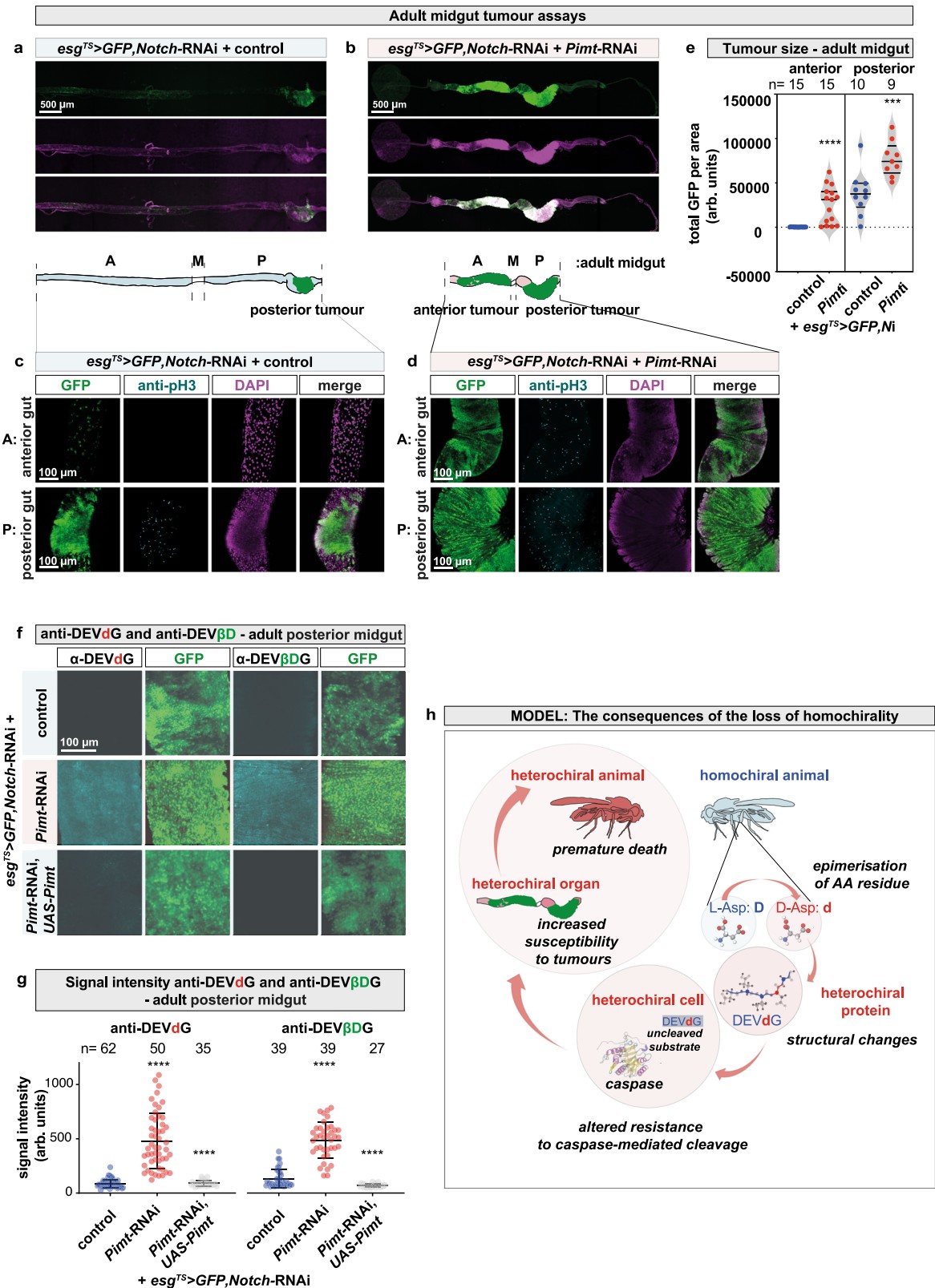

**Adult midgut tumour assays**

**a** esg^TS>GFP,Notch-RNAi + control
**b** esg^TS>GFP,Notch-RNAi + Pimt-RNAi
**e** Tumour size - adult midgut

**c** esg^TS>GFP,Notch-RNAi + control
**d** esg^TS>GFP,Notch-RNAi + Pimt-RNAi

**f** anti-DEVdG and anti-DEVβD - adult posterior midgut

**h** MODEL: The consequences of the loss of homochirality

**g** Signal intensity anti-DEVdG and anti-DEVβDG - adult posterior midgut

dsDNA donor templates for homology-directed repair (HDR) and is designed for replacing a targeted locus with a 50-bp attP phage recombination site and is positively marked with a removable (floxed) 3XP3-dsRed construct for screening. *Pimt^nulll* mutant gRNAs, were cloned into pCFD5 vector between EagI and XbaI cloning sites by gene synthesis (Genscript). pCFD5 is a vector for expressing gRNAs under the control of the RNA pol III promoter, U6:3. The fly line *yw;*

*attP40{nos-Cas9}/CyO* (Bestgene) in a *yw* background was used (the original *nos-Cas9* line is from NIG-FLY stock in a *y2 cho2 v1* background) for generating *Pimt^nulll* mutant lines[100].

**Generation of Pimt knock-in (*Pimt^wt*, *Pimt^S60Q*, *Pimt-Gal4*) lines**
The entire *Pimt^wt* or *Pimt^KI S60Q* sequences were cloned into RIV FRT vector between BglII and XhoI cloning sites by gene synthesis

**Fig. 6 | Loss of proteome homochirality enhances susceptibility to tumour formation.** Confocal images of dissected female guts. Schematic representation indicates the different regions of the adult female gut: A anterior-, M middle-, and P posterior midguts of animals in which an internal control (**a** and **c**) or *Pimt* (**b** and **d**) was knocked down simultaneously with *Notch* using *esg^TS* (*escargot* thermo-sensitive) in progenitor cells (PCs), which are marked with GFP. **e** Quantification of the area of GFP-marked tumours in female anterior (left) and posterior (right) guts of homochiral control versus heterochiral *Pimt*-RNAi animals. Experiments were repeated at least three independent times, *n* number of animals. Values are presented as average ± standard deviation (S.D.). *P*-values from two-sided Mann–Whitney U-test are ****p < 0.0001, and ***p = 0.0009. **f** Immunostaining of 20 days old female posterior gut with anti-DEV*d*G and anti-DEV*βD*G antibodies. **g** Quantification of signal intensities of anti-DEV*d*G and anti-DEV*βD*G stainings from **f**, measured by ImageJ. All quantitative analyses were carried out on unadjusted raw images. Experiments were repeated at least three independent times, *n* number of animals. Values are presented as average ± standard deviation (S.D.). *P*-values from two-sided Mann–Whitney U-test are ****p < 0.0001. Source data are provided as a Source Data file for **e** and **g**. **h** Model showing the cascading effect of loss of protein homochirality, with the active preservation of proteome homochirality being critical for maintenance of homoeostasis. Heterochirality build-up has large negative consequences across the biological scales causing severe impairment at the molecular, cellular, organ, and organism levels.

(Genscript). Pimt Gal4 knock-in was cloned into RIV FRT vector between NotI and NheI cloning sites by gene synthesis (Genscript). The *Pimt^nullI* line was used for injection of *Pimt^wt*, *Pimt^KI S60Q*, *Pimt-Gal4* constructs and knock-ins were inserted into the attP site of *Pimt^nullI* locus.

## Fly husbandry

Fly stocks were reared on a standard cornmeal/agar diet (8.25% cornmeal, 3.4% yeast, 1% agar, supplemented with 0.375% Moldex). All experimental flies were kept at room temperature (RT) or at 29 °C on a 12 h light/dark cycle. Flies crosses were set up and kept for 3 days at RT. Flies were transferred to fresh vials every day, and fly density was kept to a maximum of 15 flies per vial. For induction of *esg^TS > GFP,Notch*-RNAi tumours, crosses were kept at 18 °C. After eclosion, female imagoes were placed at 29 °C for 15 days prior to dissection. For immunohistochemistry experiments, female imagoes were kept for 20 days prior to dissection.

## Generation of chiral-selective antibodies

Polyclonal antibodies were produced in rabbits against the following synthetic oligopeptide epitopes (Cambridge Research Biochemicals Limited, UK): [C]-GG-DEV-[L-aspartate]-G-amide for anti-DEV**D**G, [C]-GG-DEV-[iso-L-aspartate]-G-amide for anti-DEV**βD**G, [C]-GG-DEV-[D-aspartate]-G-amide for anti-DEV*d*G, and [C]-GG-DEV-[iso-D-aspartate]-G-amide anti-DEV**βd**G. For affinity purification, two conditions were used for antibody elution, a low pH (pH 2) Glycine was used for the first elution, followed by a high pH (pH 10.5) Triethylamine TEA for the second elution. Both eluates were neutralised and then dialysed in PBS.

## Pimt colorimetric continuous enzyme coupled assay

SAM Methyltransferase colorimetric assay (Merck) was used to measure the activity of purified recombinant human Pimt (EC 2.1.1.77, MBS144034, Gentaur) on different target substrates at 37 °C. Pimt (1 μM final concentration) enzymatic activity was followed spectrophotometrically by the increase in absorbance at 510 nm using DeNovix DS-11 Series microvolume Spectrophotometer/Fluorometer. Assay was performed on different homo- and heterochiral synthetic substrates Ac-KRKGDEVDGVDEVAK-amide (Peptide 1), Ac-KRKGDEV-(iso-L-Asp)-GVDEVAK-amide (Peptide 2) (both of which HPLC purity is >90%), at 100 μM final concentration. The peptide's sequence is identical and correspond to the PARP1 fragment which is cleaved by Dcp-1 in the in vivo cleavage assay.

## Enzyme-linked immunosorbent assay (ELISA)

ELISA assay was used for cross-reactivity screening of chiral-selective antibodies and four different synthetic epitopes. The antigens were chemically synthesised oligopeptides as follows: [C]-GG-DEV-[L-Asp]-G-amide, [C]-GG-DEV-[D-Asp]-G-amide, [C]-GG-DEV-[iso-L-aspartate]-G-amide, [C]-GG-DEV-[iso-D-aspartate]-G-amide. ELISA plates were coated with free peptide antigens at 10 μg/ml in PBS pH 7.4. 2 μL of purified antibody was added to the plate in duplicate. Samples were serially diluted from 1/50 to 1/51.200 in blocking buffer PBS-T (PBS/0.01% Tween-20). Alkaline Phosphatase conjugated anti-rabbit IgG specific antibody (1:500) and PNPP (p-nitrophenyl phosphate) substrate was added, and absorbance was determined at 405 nm on an Epoch Microplate Spectrophotometer (Biotek).

## ChiCAS: Chiral-selective in vitro DEVD-ase assay

The chiral-specific in vitro caspase cleavage (DEVDase) assay is based on the release by Caspase-3 of a fluorescent moiety (AMC) from the hydrolysis of a synthetic fluorogenic pentapeptide probe Ac-DEVD-AMC (acetyl-{L-Asp}(P4)-{L-Glu}(P3)-{L-Val}(P2)-{L-Asp}(P1)−7-amido-4-methylcoumarin). The following homochiral and heterochiral fluorogenic substrates were chemically synthesised: Ac-DEV-(l-aspartate)-AMC (positive control), Ac-DEV-(iso-L-aspartate)-AMC, Ac-DEV-(D-aspartate)-AMC, Ac-DEV-(iso-D-aspartate)-AMC, Ac-DEVG-AMC and Ac-DEV-(L-alanine)-AMC (negative control) and Ac-(D-aspartate)-(D-glutamate)-(D-valine)-(D-aspartate)-AMC (custom peptide synthesis by GenScript). All synthetic peptides were dissolved in DMSO and stored at −80 °C. Reactions were performed as described below. Ac-DEVx-AMC substrates were used in a final concentration of 20 μM in assay buffer (50 mM HEPES, 100 mM NaCl, 0,1% CHAPS, 1 mM EDTA, 10% Glycerol, 10 mM DTT - immediately added before use). Assay was performed with recombinant human active Caspase-3 (556471, BD Biosciences). Reactions were performed in 96 well microplates (100 μl/reaction) and caspase activity was monitored spectrophotometrically by the increase in fluorescence at 460 nm using Synergy 4 BioTek fluorescence absorbance microplate reader and Gen5 program.

## ChiFCA assay: genetically encoded chiral-specific in vivo fluorometric caspase activity assay

*UASp-tDcp-1* (Caspase-3 functional homologue) and *UAS-CD8::PARP1::Venus* (caspase substrate) were simultaneously expressed under the control of *nub-Gal4* (for salivary gland) or *Act5C-Gal4* (adult gut). Dissected and fixed larval salivary glands and adult midguts were stained with anti-DEV*d*G antibody and monitored at 25x magnification using a Zeiss 780 confocal laser scanning microscope. After imaging, channels were split and a single Z-stack was analysed. To calculate the proportion of membrane localised and cytosolic GFP signal intensities, lines were drawn between adjacent cells and inside the cytosol and signal intensity was measured using ImageJ. Data were collected from at least three independent experiments, and a minimum of 12 salivary gland or gut cells per genotype and/or condition were analysed. The ratio of cytosolic and membrane GFP signal intensities was quantified and expressed as ratio signal intensities (±S.D.).

## Immunohistochemistry

Larval tissues were stained using standard immunohistochemical procedures. Larval or imaginal organs were dissected in PBS, fixed at room temperature for 20 min in 3.7% formaldehyde/PBS, and washed in 2% Triton-X100/PBS. Embryos were kept in 3% bleach for 7 min for dechorionation, washed, completely dried and then fixed at room temperature for 20 min in 3.7% formaldehyde/heptane (1:1). Formaldehyde and heptane were gradually replaced with methanol. All

subsequent incubations were performed in 0.1% Tween20/PBS (embryos and gut) or in 0.2% Triton X-100/PBS (organs) at 4 °C.

Samples were mounted in Vectashield containing DAPI (Vector Labs). The following primary antibodies were used: Chicken anti-GFP (1:2000, ab13790, Abcam), Rabbit anti-Phospho-Histone H3 Ser10 (1:500, 9701L, Cell Signalling Technology), Rabbit anti-[C]-GG-DEV-[L-aspartate]-G-amide or anti-DEVDG (1:200, this work), Rabbit anti-[C]-GG-DEV-{iso-L-aspartate}-G-amide or DEV*βD*G (1:200, this work), Rabbit anti-[C]-GG-DEV-{D-aspartate}-G-amide or DEV*d*G (1:200, this work), and Rabbit anti-[C]-GG-DEV-{iso-D-aspartate}-G-amide or DEV*βd*G (1:200, this work). Fluorescent secondary antibodies: Donkey anti-Rabbit Alexa Fluor 594 (711-585-152), Donkey anti-Rabbit Alexa Fluor 488 (711-546-152), Donkey anti-Chicken Alexa Fluor 488 (703-545-155) (all 1:2000, Jackson ImmunoResearch) and Goat anti-Mouse Alexa Fluor 647 (1:2000, A32728, Invitrogen). Each experiment was performed at least three independent times.

### Quantification of the proportion of GFP-positive areas

Dissected adult guts discs were imaged at 10x magnification using a Zeiss 780 confocal laser scanning microscope. Four Z-stacks were taken for each disc. After imaging, channels were split, and maximum Z-projection was analysed. To calculate the proportion of GFP-positive areas per adult gut, a line was drawn around the total gut area using DAPI channel images and measured using ImageJ. Then the sum area of GFP-positive cells was measured using the GFP channel. The threshold was adjusted with the IsoData auto threshold algorithm to subtract background and the area above the threshold was analysed. Data were collected from at least three independent experiments, and a minimum of 10 guts per genotype and/or condition were analysed. The relative occupancy of GFP positive cells was quantified and expressed as the proportion of GFP positive area per gut area (±S.D.).

### Western blot analysis

Proteomes were purified from dissected guts of whole adult flies (100 guts/sample) using a lysis buffer (20 mM TrisHCl pH 7.8, 150 mM NaCl, 0.5 mM EDTA, 1 mM protease inhibitor), incubated on ice for one hour (vortexed each 10 min) and centrifuged for 20 min at 20.000 g at 4 °C with Eppendorf 5424 R benchtop centrifuge. Supernatants were carefully removed, concentrations were measured with DeNovix DS-11 Series microvolume Spectrophotometer/Fluorometer, and equilibrated. Proteins were separated using NuPage™ 12% Bis-Tris (NP0341BOX Invitrogen) gels, using non-denaturing conditions. Proteins were blotted onto Immobilon-P transfer membrane (IPVH00010, Millipore), and blots were blocked with 5% BSA/TBST. Blots were incubated in Rabbit anti-DEV*d*G (1:1000), anti-DEV*βD*G (rabbit, 1:1000) or Rabbit anti-α-Tubulin (1:1000, 2144 S, Cell Signalling Technology) primary antibodies over-night at 4 °C and in ECL Donkey anti-Rabbit-HRP (1:2000, Amersham NA934-1ML) secondary antibody for 1 h at RT. Blots were developed with Immobilon Western chemiluminescent HRP substrate (WBKLS05000, Millipore) with ImageQuant™ LAS 4000 biomolecular imager. Anti-α-Tubulin loading control was used for calculating total lane density and signal intensities of protein bands was measured with ImageJ. All samples derive from the same experiment and gels/blots were processed in parallel. Source data are provided as a Source Data file. The normalised signal intensitiy of protein bands was quantified and expressed as average relative density (±S.D.).

### Statistics and data presentation

All statistical analyses were carried out using Microsoft GraphPad Prism 9.2.0 (283) and Microsoft Excel (version 16.16.27). Confocal images belonging to the same experiment and displayed together were acquired using the same settings. ZEISS ZEN software (2019) was used for data collection/capturing images. For visualisation, the same level and channel adjustments were applied using ImageJ (2015). Of note, all quantitative analyses were carried out on unadjusted raw images or maximum projections. Comparisons between two genotypes/conditions were analysed with Mann-Whitney nonparametric two-tailed rank U-test. Values are presented as average ± standard deviation (S.D.). P-values from Mann−Whitney U-test are ****$p < 0.0001$; $0.0001 < ***p < 0.001$; $0.001 < **p < 0.01$; $0.01 < *p < 0.05$; and not significant ns $0.05 < p$. For age-specific survivorship experiments, comparison of survival curves was done with Log-rank (Mantel-Cox) test. P values from Log-rank test are ****$p < 0.0001$; $0.0001 < ***p < 0.001$; $0.001 < **p < 0.01$; $0.01 < *p < 0.05$; and not significant ns $0.05 < p$. Multiple comparisons were done with One-way ANOVA test. P values from One-way ANOVA tests are ****$p < 0.0001$; $0.0001 < ***p < 0.001$; $0.001 < **p < 0.01$; $0.01 < *p < 0.05$; and not significant ns $0.05 < p$. Data from western blotting were analysed with unpaired, two-tailed Student's t-test. P values from unpaired, two-tailed Student's t-tests are ****$p < 0.0001$; $0.0001 < ***p < 0.001$; $0.001 < **p < 0.01$; $0.01 < *p < 0.05$; and not significant ns $0.05 < p$. For data presentation, Adobe Illustrator 2021, Microsoft Word (version 16.16.27) and EndNote Cite While You Write were used.

### Preparations of thin film samples

Thin films of L-, and D-Asp were prepared from the sublimations of corresponding sample powders (≥99%, Sigma) in vapour deposition chamber (ASAHI-SEISAKUSYO, Japan), under a high vacuum (around $10^{-5}$ Torr). The thickness of thin films was monitored using the 6 MHz quartz crystal (Inficon, Japan). The deposition started at 152 °C, and the thickness gradually increases to 68 Å at around 233 °C. The thin film of L-Asp with thickness of 43 Å was obtained in the same manner.

### Far-UV synchrotron radiation circular dichroism (SRCD)

Amino acid samples were analysed at beamline BL-12 at HISOR Synchrotron[101]. Amino acid 43 Å thick film of L-aspartic acid and 68 Å thick film of D-aspartic acid sublimated on $CaF_2$ crystal disks were then mounted in the VUVCD spectrometer under nitrogen purging at 25 °C. VUVCD spectra were collected between 165–250 nm at 1 nm bandwidth, 16-s time constant and 4 nm min$^{-1}$ scan speed averaging 5 accumulations. Noise and inaccuracies of the film thickness contribute to a 5% error within the reproducible measured ellipticity.

Circular Dichroism spectra of synthetic homochiral and heterochiral oligopeptides were collected at the Synchrotron-Radiation Circular Dichroism (SRCD) end-station on DISCO beamline at the SOLEIL Synchrotron under project 20181617 and 20180582. The investigated peptides were peptide antigens that we used to produce iso-L-, and D-specific antibodies described. Samples were deposited between two CaF2 windows, 22 μm was chosen as the optimised pathlength for the given concentrations. Triplet spectra were acquired of peptides and their respective baselines using 1 nm bandwidth, with a 1.2 s integration time collecting every 1 nm from 260–175 nm, then averaged and smoothed (5 nm using Savitzky-Golay algorithm). Peptide spectra were baseline subtracted normalised with (+) camphor sulfonic acid standard and standardised to Δε as described previously[102,103]. Data-analyses including averaging, baseline subtraction, smoothing, and scaling were carried out with CDtool software[104].

### Reporting summary

Further information on research design is available in the Nature Portfolio Reporting Summary linked to this article.

## Data availability

All data is available in the main text or the supplementary data. Source data are provided with this paper. Materials generated for the study are available from the corresponding author on reasonable request.

The MD simulation data (starting molecular structures, topologies, trajectories) along with simulation results (timelines, summaries and residual decomposition of binding energies, distance profiles, and interaction energies) have been deposited in the Zenodo database and is freely available under accession code: https://doi.org/10.5281/zenodo.7199808 [https://zenodo.org/record/7199808#.Y0wKKOzMJ_Q].

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

## Acknowledgements

We thank Masayuki Miura, Joaquín de Navascués, Eli Arama and Kim McCall for reagents; the Bloomington Drosophila Stock Centre and Vienna Drosophila RNAi Centre (VDRC) for providing *Drosophila* fly lines; the Irish Centre for High-End Computing (ICHEC) for supercomputing resources, the iBV PRISM platform; the French national synchrotron facility - Synchrotron SOLEIL; Pascal Meier, Tencho Tenev, Florence Besse, Maximilian Fürthauer, Maria Duca for comments; Jean-Marc Gambaudo and Sylvie Mellet for their support, and members of the SN laboratory for discussions. Funding: Université Côte d'Azur (UCA), IDEX- Initiative d'excellence Grant (A.B.), SOLEIL Synchrotron grant, Project number 20181617 (A.B.), SOLEIL Synchrotron grant, Project number 20180582 (A.B.), Science Foundation Ireland (SFI) grant, award number 12/RC/2275_P2 (DT), ERC starting grant CellSex, Grant number: ERC-2019-STG#850934 (B.H.). Work in SN laboratory is supported by Agence Nationale pour la Recherche (ANR-17-CE13-0024; ANR-20-CE13-0004), Fondation pour la Recherche Médicale (FRM; EQU201903007825), Université Côte d'Azur (UCA), Centre National pour la Recherche Scientifique (CNRS), Institut National pour la Recherche Médicale (Inserm), LABEX SIGNALIFE (ANR-11-LABX-0028-01).

## Author contributions

Conceptualisation: A.B., S.N. Methodology: A.B., F.W., S.B., K.M. Investigation: A.B., F.W., S.B. Visualisation: A.B., S.N., S.B. Funding acquisition: A.B., S.N., B.H., D.T. Project administration: A.B., S.N. Supervision: A.B., D.T., S.N. Writing – original draft: A.B., S.N., D.T., S.B. Writing – review & editing: A.B., S.N., D.T., S.B., U.M., F.W., B.H., C.M., M.R.

## Competing interests

The authors declare no competing interests.
