## [Peer Review File · Nature Communications]

Biological effects of the loss of homochirality in a multicellular organismREVIEWER COMMENTS

Reviewer #1 (Remarks to the Author):

The article titled "Biological effects of the loss of homochirality in a multicellular organism", by Agnes Banreti et al, emphasizes the absolute requirement of homochirality in proteins for cell survival. The work presented is a useful addition to our understanding of the criticality of enforcement of homochirality across life forms, with the focus on *Drosophila*. However, the article suffers from serious presentation issues and also does not seem to add or highlight any new information to understand the importance larger issue of protein homochirality. Even if it has, the authors have not spelt out what new information this work has come up with when compared to previous literature on this issue. For example, *pimt* gene is well studied in both prokaryotes (bacteria, (Yin et al, JBC, 2018)), and eukaryotes (mice (Kim et al, PNAS, 1997; Young et al, JBC, 2005); *C. elegans* (Kagan et al, Archives of Biochemistry and Biophysics, 1997); plants (Wei et al, Plant Molecular Biology, 2015; Gosh et al, JBC, 2019)). These studies have already shown the role of *pimt* in growth, longevity, epithelial to mesenchymal transition in lung adenocarcinoma and faster migration and invasion of glioblastoma cell lines. The authors should summarize these results and then talk about the unique features of their work!

Overall, the claims must be toned down significantly and the manuscript must be re-written in a major way by appropriately summarizing previous literature, discussion of unique results etc.

Major concerns:

- Use of "loss of homochirality" in title is an overstatement. The manuscript is talking about just one amino acid out of 19 L-forms. Loss of homochirality is expected to be lethal and hence there are multiple machineries that ensure chiral fidelity during protein biosynthesis. Authors have used contradictory statements in abstract itself. They state that "Homochirality is a fundamental feature of all known forms of life" in the very first line of the abstract but also state that "essential to maintain homeostasis of multicellular organisms" in last line. "Multicellular" usage suggest homochirality is required in only multicellular organisms, which is untrue. Also, enormous amount of information is available on the mechanisms preserving homochirality. D-amino acids are not "unnatural". Moreover, various other model systems are already genetically altered to study protein homochirality before this work.
- The authors generated CRISPR-based *pimt* knockout and utilised the model for confirming the formation of heterochiral peptides through antibodies and for proving the reduction of lifespan experiments. But in the other experiments, the authors utilised RNAi mediated knockdown lines of *pimt*. The authors failed to justify the usage of two different systems in different experiments.
- Why authors have not considered the peptides with highest *pimt* activity based on earlier literature (Lowenson and Clarke, JBC, 1991)? How authors arrived at using DEVDG motif as a specific site for *pimt*? How many proteins contain this site and their biological functions? This will help in explaining the observed phenotypes.
- Authors are more focused on biochemical experiments with caspase and not *pimt*. While caspase-based assays are well established. Authors need to show biochemical activity of *pimt* on DEVDG peptides.
- Line no: 24, "deleterious" is improper in this context as the organism does survive.
- "Biological effect of loss of homochirality" need to be rephrased as "partial or minimal loss of homochirality" as only Asp is studied here and not other D-amino acids that are present in significant quantity.
- Line no: 44-47 implies that no studies are done to understand the homochirality in animals which is untrue.
- Chiral-selective antibodies were already used in previous literature to detect heterochiral peptides in vivo. Therefore, first result does not seem to add any value to the previous information. It is important to mention what additional benefit it have over earlier used antibodies.
- Heterochiral peptides are already known to have distinct confirmation than homochiral ones so second result section also needs to be emphasized with regard to its novelty.
- Literature review for using "DEVDG" motif, its implications must be written.

- Literature review on pimt Knock-out studies on other model systems must also be mentioned and discussed.
- Western blot-based quantification is required to confirm the results reported in result number 1b, as the confocal image suggests intensity changes.
- In result 5, aggregation of protein is reported, control panel for the mentioned result is necessary to show that it is pimt specific observation.
- Result 6 suggests the accumulation of DEVDG peptides in the tumour regions. This result did not explore the presence of other heterochiral peptides.
- How authors arrived at the possibility of checking tumor generation in pimt knock out flies is not clear?
- Heterochiral peptides are detected in aged animals including humans. Can authors comment why these peptides got accumulated even in the presence of pimt enzyme?

Minor concerns:

- 1) Line no: 31-34- give references
- 2) In line no 35: Please replace "chiral proofreading by ribosome" with "various chiral checkpoints like aminoacyl-tRNA synthetase, elongation factor thermos unstable, ribosome and chiral proofreading by D-aminoacyl-tRNA deacylase during protein synthesis"
- 3) Figs (3c-e) requires further explanation in the main text. How "DEVdG" variant escapes the catalytic dyad pocket based on distance value must be mentioned clearly.
- 4) Hyphen must be removed after homo at multiple places in the text (example: line no. 67, figure 2, line no. 211 etc.)
- 5) In figure 2, color coding between iso-L-aspartate and iso-D-aspartate is confusing.
- 6) As the work was done only in flies therefore, the word "animals" should be used appropriately in the manuscript.
- 7) Fig. 6G has alignment issues in figure legends.
- 8) Abstract: protein-bound???

Reviewer #2 (Remarks to the Author):

In this manuscript, the authors study the effect of loss of homochirality biochemically and in an intact animal, *Drosophila melanogaster*. They generated a Pimt mutant in *Drosophila*. Pimt encodes a protein that recognizes abnormal D-Asp and L-iso-Asp residues and corrects them to L-Asp. In addition, they generated four very specific antibodies which enable the detection of homochiral and heterochiral versions of the caspase cleavage site DEVDG, specifically at the P1 position. With these tools and a number of specific assays they developed, they examined the consequences of heterochirality in vivo. They show that in Pimt mutants, heterochiral epitopes accumulate. They also confirm this finding by ELISA assays. Furthermore, they show by synchrotron radiation circular dichroism spectroscopy (SRCD) that incorporation of heterochiral residues changes the structure of the affected protein. They confirm this later in the manuscript by computer modeling of caspase substrates. In chiral specific in vitro cleavage assays where they assay fluorescent homochiral and heterochiral tetrapeptide substrates, caspase-3 was only able to cleave the homochiral substrate. Computer simulations of these substrates suggests structural changes so that the caspase-3 cannot bind the heterochiral substrates, hence it is unable to cleave them. Consistently, in vivo a transgenic substrate (CD8-PARP-Venus) cannot be cleaved by the caspase Dcp-1 in a Pimt mutant/RNAi background. Finally, the authors examine the phenotypic consequences of loss of homochirality in Pimt mutant flies. These mutants display reduced life span, contain protein aggregates, form melanotic tumors and display an enhanced susceptibility to tumor formation.

Overall, this is a very interesting manuscript. The data are convincing and statistically verified. All the experiments are well explained and often graphically illustrated. The paper is also well written.

I enjoyed reviewing this manuscript (and I don't say this often). Everybody knows from their biochemistry classes, that amino acids are incorporated into proteins in their L form. But at least I never thought about the consequences if the D form is used. The authors here generated genetic (the Pimt mutant) and diagnostic (the antibodies) tools to address this question. They specifically looked at caspases and their inability to cleave heterochiral substrates, but it is more than likely that many other biological processes are affected and give cause to disease. The antibodies generated by the authors will be of significant value for diagnostic purposes. There is a lot to be done, and the authors opened the door to it.

I really only have a few formal comments/questions to further improve this manuscript:

1. In line 75, the authors report that the anti-DEVDG antibody had no reactivity in homochiral animals. They then explain, that the homochiral epitopes are not exposed at the surface in target proteins (line 77). How then do caspases recognize, bind and cleave their substrates under normal conditions? Isn't it also possible that this antibody does not work in immunofluorescent assays?

2. They authors show that a significant fraction of the Pimt mutants have melanotic tumors. Certain Toll and Jak/Stat pathway mutants also form melanotic tumors. Could the observation that Pimt mutants form these tumors mean that loss of Pimt also deregulates Toll and/or Jak/Stat signaling?

3. In lines 134/135, the authors write that "Pimt-RNAi cells had lowered caspase activity". However, I think that is a misinterpretation. The caspase activity is likely unchanged, the heterochiral CD8-PARP-Venus reporter is just not cleaved anymore because of the structural change.

4. Introduction and Discussion are quite short. It would be nice if the authors can extend these a little.

Reviewer #3 (Remarks to the Author):

D-amino acids have aroused interest in the past years since they have been detected in age-related processes, in particular D-aspartic acid residues in eye lens and brain.

To study the effect of D-aspartyl residues the authors here present a plethora of experiments. In an in vitro assay they studied the effect of exchanging a critical Asp residue in a caspase-3 cleavage assay. MD calculations undermined that the D-variant is structurally hindered to enter the catalytic pocket of caspase-3. The authors also established a Drosophila model to study the effect of altered protein heterochiral repair by creating a Pimt null mutant. Pimt recognizes and converts D-aspartyl and iso-L aspartyl residues to the native L-aspartyl form within proteins. Pimt knock out flies died prematurely, but normal life span could be recovered by a Pimt wt knockin. Furthermore, they observed that progenitor cells (PCs) of the thermosensitive Notch-induced tumour model system NotchTS-RNAi flies developed tumours when Pimt was knocked out.

The experiments are well-conceived and cover a wide range of methods. Introduction and Discussion, however, give a less diligent impression. Both are very brief and contain commonplace arguments, but do not cite specific groundwork from the literature. Especially what is known on the effect of D-aspartic acid in proteins and how they are incorporated needs to be covered more extensively in the introduction. I suggest that in order to write a more thorough discussion, the authors report on established literature, e.g. what is known on reduction of lifespan by D-amino acids, and discuss what is new and exciting in their work compared to previous results and how their experiments expand previous knowledge.

A thorough revision of abbreviations and namings would also help. For example, Pimtn1#1/ Pimtn2#1 is not explained, neither is Pimtn1 #1/w1118. Labeling of antibodies is not always consistent in main text, Figure and Figure legend. E.g. the iso-L-form is denoted as DEVisoDG (text), anti-DEVisoDG (legend) and DEVbetaD (Figure). This is confusing for people not familiar with this sort of experiments.

I must admit that I was a bit disappointed after reading title and abstract to find out that the authors only examine the effect of D-aspartic acid and not heterochirality or loss of homochirality in more general terms. Thus, I strongly advise to use a more moderate title and also adjust the abstract accordingly. The sentence "To address this question, we genetically altered protein homochirality in *Drosophila* and developed chiral-selective assays to detect protein-bound D-amino acids and assess their functional significance." is misleading in my eyes because you did not test any other amino acid apart from D-aspartic acid.

Response to reviewers

We thank the reviewers for their time in examining our manuscript and for providing their valuable comments. We consider their opinion and suggestions to be crucial and have addressed them in full in the revised manuscript. In the following section, we have clarified and expanded on all points raised and discussed in detail the corresponding changes made to the revised manuscript which are highlighted in yellow.

Reviewer: 1

The article titled “Biological effects of the loss of homochirality in a multicellular organism”, by Agnes Banreti et al, emphasizes the absolute requirement of homochirality in proteins for cell survival. The work presented is a useful addition to our understanding of the criticality of enforcement of homochirality across life forms, with the focus on *Drosophila*. However, the article suffers from serious presentation issues and also does not seem to add or highlight any new information to understand the importance larger issue of protein homochirality. Even if it has, the authors have not spelt out what new information this work has come up with when compared to previous literature on this issue. For example, *pimt* gene is well studied in both prokaryotes (bacteria, (Yin et al, JBC, 2018)), and eukaryotes (mice (Kim et al, PNAS, 1997; Young et al, JBC, 2005); *C. elegans* (Kagan et al, Archives of Biochemistry and Biophysics, 1997); plants (Wei et al, Plant Molecular Biology, 2015; Gosh et al, JBC, 2019)). These studies have already shown the role of *pimt* in growth, longevity, epithelial to mesenchymal transition in lung adenocarcinoma and faster migration and invasion of glioblastoma cell lines. The authors should summarize these results and then talk about the unique features of their work! Overall, the claims must be toned down significantly and the manuscript must be re-written in a major way by appropriately summarizing previous literature, discussion of unique results etc.

We thank the Reviewer for the constructive comments and suggestions. To address the reviewer’s point regarding provision of more detail and context for non-expert readers (also raised by the other referees) we now provide a more complete introduction and conclusion. We emphasise that our aim was to use loss of *pimt* as a tool to understand the consequences of aspartate heterochirality at molecular, cellular, tissue and organism levels. Our main focus was not to investigate the *Pimt* enzyme/gene in detail, as it has been characterised in other systems as pointed out by the Reviewer. Thus, we focused on potential cellular consequences and found caspases – which are key aspartate-specific endopeptidases regulating a wide range of cellular processes – to be physiological targets of *pimt* loss of function. Substrate cleavage by caspases is frequently deregulated in malfunctioning cells and consequently considered one of the hallmarks of cancer. We therefore investigated whether susceptibility to tumour formation is altered in heterochiral animals. Accordingly, in the first version of the article we introduced caspases, their targets and apoptosis to direct the reader’s focus on this subject rather than on *Pimt* (which we aimed to use only as a genetic tool). However, we fully agree this brings imbalance to the work and we now added an extensive literature summary about *Pimt*, as suggested.

Major concerns:

- Use of “loss of homochirality” in title is an overstatement. The manuscript is talking about just one amino acid out of 19 L-forms.

Based on earlier studies showing that aspartate is the most susceptible to racemisation among all amino acids, it is the amino acid which had been predominantly found in most aged human tissues. Our understanding is that despite having 19 proteinogenic amino acids, the main cause of proteome heterochirality is the epimerisation and isomerisation of aspartate in *Drosophila*. In the new version of the manuscript, we spelled this out more clearly (please see page 4, lines 81-85).

Loss of homochirality is expected to be lethal and hence there are multiple machineries that ensure chiral fidelity during protein biosynthesis. Authors have used contradictory statements in abstract itself. They state that “Homochirality is a fundamental feature of all known forms of life” in the very first line of the abstract but also state that “essential to maintain homeostasis of multicellular organisms” in last line. “Multicellular” usage suggest homochirality is required in only multicellular organisms, which is untrue.

We thank the Reviewer for pointing out this contradiction, we removed “multicellular” from the sentence.

Also, enormous amount of information is available on the mechanisms preserving homochirality.

We have now included an extensive literature summary about mechanisms preserving homochirality (please see page 2, lines 35-38 and page 3, lines 48-51). Our focus was not to study the mechanisms maintaining homochirality but rather the mechanisms underlying the biological and pathophysiological consequences of homochirality loss.

D-amino acids are not “unnatural”.

We thank the Reviewer for pointing out this poor choice of words, and are sorry for incorrectly using “unnatural” for d-amino acids. We are aware that d-amino acids are not unnatural but this term has been frequently -and indeed incorrectly- used in the literature. We have corrected this in our manuscript to non-L- α -amino acids (page 2, line 19; page 4, line 73; page 6, line 129 and page 11, line 246).

Moreover, various other model systems are already genetically altered to study protein homochirality before this work.

Indeed, various other model systems are already genetically altered to study the effect of Pimt activity loss. However, less is known about whether these effects are due to the loss of Pimt activity itself (potentially independent of its activity on chirality maintenance) or are direct consequences of (aspartate) heterochirality. Moreover, studying the molecular mechanisms underlying the biological effects of aspartate heterochirality was not feasible *in vivo*. We used *Drosophila* as a powerful genetic model to alter aspartate homochirality, in which we established a genetically encoded chiral-sensitive cleavage assay as well as a set of complementary chiral-selective antibodies that make it possible – to our knowledge for the first time – to study the dynamics of chirality changes *in vivo* and *in situ*.

The authors generated CRISPR-based pimt knockout and utilised the model for confirming the formation of heterochiral peptides through antibodies and for proving the reduction of lifespan experiments. But in the other experiments, the authors utilised RNAi mediated knockdown lines of pimt. The authors failed to justify the usage of two different systems in different experiments.

It is standard practice to use different ways of altering a gene's function. The two genetic alterations are considered equivalent and complementary and in fact help reinforce our genetic data. In particular, the use of RNAi instead of a mutant helps generating graded phenotypes that are useful to understand the gene function, as well as tissue-specific and spatio-temporal loss of function. Also, it makes some difficult genetic experiments easier to perform (e.g., generating loss of function clones in a complex genetic context to test cell autonomy, like in Figure 6).

Why authors have not considered the peptides with highest pimt activity based on earlier literature (Lowenson and Clarke, JBC, 1991)? How authors arrived at using DEVDG motif as a specific site for pimt? How many proteins contain this site and their biological functions? This will help in explaining the observed phenotypes.

The rationale behind focusing on caspase target sequences was that aspartate has the highest epimerisation rate among all proteinogenic amino acids, specifically when followed with a small amino acid like glycine, serine or alanine, which is the case for most known caspase consensus cleavage sites (Poreba et al. 2013, PMID: 23788633; Julien and Wells 2017, PMID: 28498362; Douglas R Green 2022, PMID: 35232877). Besides Granzyme-B-, caspases are the only group of enzymes that are able to specifically cleave following aspartate residues. We thus investigated whether the incorporation of non-L- α -amino acids indeed makes target peptides resistant to proteolytic cleavage and as a consequence, leads to accumulation in the cell. The DEVDG sequence is a well-known caspase target consensus sequence, occurring in various proteins (in *Drosophila* 21 proteins shows 100% identity and more than 100 proteins shows 80% identity with the sequence) and is evolutionary conserved. With partially redundant sequences DXXD/G (or S), hundreds of substrates for caspase-3 have been identified, and investigators believe there could be a thousand or more (Douglas R Green 2022, PMID: 35232877). Caspase target proteins are cleaved by caspases upon the activation of the apoptotic pathway. While caspases have several non-apoptotic functions, one major function is to execute damaged or unwanted cells, hence caspases represent interesting physiological 'hot spots' for altered aspartate heterochirality.

- Authors are more focused on biochemical experiments with caspase and not pimt. While caspase-based assays are well established. Authors need to show biochemical activity of pimt on DEVDG peptides.

We thank the Reviewer for suggesting this important experiment. We have now added new results showing that Pimt enzyme indeed acts on the DEVDG motif (please see figure 1d on page 19 and page 20, lines 576-580).

- Line no: 24, "deleterious" is improper in this context as the organism does survive.

We initially used “deleterious” as a synonym to ‘harmful’ as is often used in the literature. To avoid any incorrect grammatical use, we have replaced it with ‘harmful/detrimental’ in the introduction (page 2, lines 33 and 35) and to “large negative” in the Model (page 30, line 674).

- “Biological effect of loss of homochirality” need to be rephrased as “partial or minimal loss of homochirality” as only Asp is studied here and not other D-amino acids that are present in significant quantity.

In our understanding, loss of homochirality does not mean that all or most of the proteinogenic amino acids suffer from stereoinversions. It is known that aspartate – due to its specific chemical structure – is the most prone to epimerisation in protein bound form among all proteinogenic amino acids and is the one that has been found previously in various aged human tissues. Any alteration from homochirality means that the animal is heterochiral. We have now modified our text and emphasised the predominant relevance of aspartate in proteome heterochirality (page 4, lines 81-85 and page 10, lines 218-221).

- Line no: 44-47 implies that no studies are done to understand the homochirality in animals which is untrue.

We fully agree that animal models have previously been used to investigate the effect of Pimt loss of function. However, none of the studies have investigated the mechanism by which heterochirality leads to various phenotypic effects. Previous work either have shown *in vivo* and *in situ* that the accumulation of D-Asp and iso-L-Asp residues and not simply the lack of Pimt activity causes the observed phenotypes. To our knowledge, our work is the first to show from molecular through cellular, organ and organism level that the observed phenotypic outcomes are indeed the direct consequences of aspartate heterochirality.

- Chiral-selective antibodies were already used in previous literature to detect heterochiral peptides *in vivo*. Therefore, first result does not seem to add any value to the previous information. It is important to mention what additional benefit it have over earlier used antibodies.

Iso-L-Asp (Brady et al. 1999, PMID: 9918902; Shimizu et al. 2000, PMID: 11032409) and a 3R (iso-D-Asp)-specific (Fujii et al. 2000, PMID: 10706893) antibodies were used to detect heterochiral proteins of aged human tissues (or urine samples) and no heterochiral-specific antibodies were tested *in vivo* like in our study. These previously described antibodies were generated against a specific sequence of a single protein. In contrast, our antibodies are recognising a motif which occurs in various proteins and thus make them a powerful tool to trace dynamic chirality changes at the proteome level. Furthermore, in previous studies, only a single antibody was raised against an epitope where one specific amino acid was altered to non-L-form, and not a set of four antibodies with all possible four L- and non-L-amino acid forms as it is presented in our study. Hence, previously no cross-reactivity against the three other chiral forms of aspartate could be addressed. The fact that we used our antibodies with genetically engineered animals overexpressing the target epitope make them an even more powerful tool to dynamically trace chirality changes *in vivo*.

- Heterochiral peptides are already known to have distinct confirmation than homochiral ones so second result section also needs to be emphasized with regard to its novelty.

The novelty of our study is that we show that conformational changes related to isomerisation and racemisation of our specific motif have biological consequences. We show that the homochiral peptide structure is perturbed by chirality changes and the consequent conformational changes indeed have biological relevance: they alter the antigenicity of the epitope as well as make the peptide motif resist proteolysis by caspases. Furthermore, these altered (isomerised and epimerised forms) of the epitopes occur naturally. To our knowledge, these have not been shown before experimentally.

- Literature review for using “DEVDG” motif, its implications must be written.

Aspartyl residues are among all amino acids the most susceptible to isomerisation and racemisation. Comparison of the primary sequences surrounding isoAsp residues reveals a strong tendency for a glycine residue to be situated at the N+1 position, consistent with an increased propensity for succinimide formation at these sites (Stephenson and Clarke 1989, PMID: 2703484; Teshima et al. 1991, PMID: 2018763; Bischoffet al. 1993, PMID: 8422378). Since the family of Caspases – the main executors of apoptosis – are among the very few and specific endopeptidases which are strictly cleaving following aspartyl residues, we aimed to investigate the physiological consequences of aspartate homochirality loss in conserved caspase cleavage sites at molecular, cellular and organism levels.

We agree with the Reviewer that it is important to provide background on this point and have added a literature review for using “DEVDG” motifs and their implications, as suggested (please see pages 4-5, lines 86-97 and page 11, lines 234-245).

- Literature review on *pimt* Knock-out studies on other model systems must also be mentioned and discussed.

We have now extended our literature review and included *Pimt* knock out studies in mice, *C. elegans* and plants in our text (see page 3, lines 54-59).

- Western blot-based quantification is required to confirm the results reported in result number 1b, as the confocal image suggests intensity changes.

In the new version of the manuscript, we now present western blot-based quantification of our antibodies (New Suppelementary Figure 2c-d on pages 47 and 48, lines 931-934) which clearly show that they indeed recognise heterochiral peptides isolated from *Pimt* KO animals.

- In result 5, aggregation of protein is reported, control panel for the mentioned result is necessary to show that it is *pimt* specific observation.

We added a control panel in Figure 5b left (shown on pages 27 and 28, line 649-650).

- Result 6 suggests the accumulation of DEVdG peptides in the tumour regions. This result did not explore the presence of other heterochiral peptides.

We have extended our study to another heterochiral peptide (DEVisoDG) which is shown in New Figure 6., panels f) and g), right (shown on pages 29 and 30, lines 666-671).

- How authors arrived at the possibility of checking tumor generation in pimt knock out flies is not clear?

The rationale behind focusing on tumour generation was that we observed melanotic tumour masses in animals lacking Pimt enzymatic activity, which we show in Figure 5c (please see pages 27 and 28, lines 651-655).

- Heterochiral peptides are detected in aged animals including humans. Can authors comment why these peptides got accumulated even in the presence of pimt enzyme?

There are several possible scenarios including that in aged animals, Pimt activity declines, not all Aspartyl residues are recognized and repaired by Pimt as well as there are other protein repair enzymes which maintain.

This is one of the most interesting and exciting questions we would be happy to investigate in future studies. Especially we aim at identifying, for example through genetic screening, new genes and enzymes which are also implicated in the maintenance of proteome homochirality. It is well known that aspartyl residues have different level of propensities to isomerise into iso-L-Asp form or to racemise into D-Asp form, depending on the upstream sequence as well as the local topology within the protein, surrounding the Asp residue. However, the level of these propensities does not necessarily correlate with their accessibility to Pimt-mediated methylation and repair. We find it very highly likely that, besides Pimt, other protein repair enzymes with different heterochiral substrate specificity might exist as well as endopeptidases which process iso-L-aspartyl and D-aspartyl sites (the latter had already been found as an orphan enzyme in mammals). Together with the scenarios proposed by the Reviewer, it is also possible that the subcellular compartmentalisation of Pimt results in the methylation and repair of mostly the cytosolic/membrane proteins while very low level of methyl-accepting activity are associated with nuclear proteins (Clare M. O'Connor, 1987, PMID: 3611066).

A likely scenario is that the age-related accumulation of iso-L-Asp and D-Asp residues in the proteome is multifactorial and the combined result of all the above-mentioned effects.

Minor concerns:

1. Line no: 31-34- give references.

A list of references were added to lines 32-35 on page 2 (original version 31-34).

2. In line no 35: Please replace “chiral proofreading by ribosome” with “various chiral checkpoints like aminoacyl-tRNA synthetase, elongation factor thermos

unstable, ribosome and chiral proofreading by D-aminoacyl-tRNA deacylase during protein synthesis”

We thank the Reviewer for suggesting this correction. This has been added in the text with an extensive list of references (please see page 2, lines 35-38).

3. Figs (3c-e) requires further explanation in the main text. How “DEVdG” variant escapes the catalytic dyad pocket based on distance value must be mentioned clearly.

We thank the reviewer for their suggestion and acknowledge the need for clarity on how the DEVdG chiral mutant escapes the catalytic pocket. To address this concern based on Fig. 3c–e in the main text, we have added an additional supplementary Figure S13 (please see pages 64-65, lines 1122-1127), where we map the timelines of distances of caspase-3 residues that make favourable contacts with DEVdG within the first ~140 ns and residues that repel DEVdG after ~140 ns of dynamics, along with corresponding snapshots during specific timepoints. We have thus added a paragraph explaining the new data under supplementary note S13 Supplementary analysis (pages 59-60, lines 1080-1093).

Further we have updated portions of the main text that refers to the newly added Fig. S13 (pages 7-8, lines 157-162).

4. Hyphen must be removed after homo at multiple places in the text (example: line no. 67, figure 2, line no. 211 etc.)

This has been corrected (p5, line 94; p22 line 590; p35 line 761 and p39, line 867). Thank you.

5. In figure 2, color coding between iso-L-aspartate and iso-D-aspartate is confusing.

We thank the Reviewer for this comment. We agree that the colour coding was confusing and changed the colour of iso-d-aspartate from light green to light yellow (please see page 21 and also pages 19 and 23).

6. As the work was done only in flies therefore, the word “animals” should be used appropriately in the manuscript.

We replaced “animals” to “Drosophila” to make it clear that the conclusions were made based only on results in flies (please see page 10, line 225, original draft page 9, line 170).

7. Fig. 6G has alignment issues in figure legends.

We could not find alignment issues in the original version of the Fig. 6g. In the new version it is represented on page 30, lines 668-671.

8. Abstract: protein-bound???

In the abstract we would like to emphasise that our study is on protein-bound amino acids rather than free L- and D-amino acids. While aspartate occurs in

iso-L-Asp and iso-D-Asp form exclusively in proteins, the L-Asp and D-Asp isoforms can be found as free amino acids across species.

Reviewer: 2

In this manuscript, the authors study the effect of loss of homochirality biochemically and in an intact animal, *Drosophila melanogaster*. They generated a Pimt mutant in *Drosophila*. Pimt encodes a protein that recognizes abnormal D-Asp and L-iso-Asp residues and corrects them to L-Asp. In addition, they generated four very specific antibodies which enable the detection of homochiral and heterochiral versions of the caspase cleavage site DEVDG, specifically at the P1 position. With these tools and a number of specific assays they developed, they examined the consequences of heterochirality in vivo. They show that in Pimt mutants, heterochiral epitopes accumulate. They also confirm this finding by ELISA assays. Furthermore, they show by synchrotron radiation circular dichroism spectroscopy (SRCD) that incorporation of heterochiral residues changes the structure of the affected protein. They confirm this later in the manuscript by computer modeling of caspase substrates. In chiral specific in vitro cleavage assays where they assay fluorescent homochiral and heterochiral tetrapeptide substrates, caspase-3 was only able to cleave the homochiral substrate. Computer simulations of these substrates suggests structural changes so that the caspase-3 cannot bind the heterochiral substrates, hence it is unable to cleave them. Consistently, in vivo a transgenic substrate (CD8-PARP-Venus) cannot be cleaved by the caspase Dcp-1 in a Pimt mutant/RNAi background. Finally, the authors examine the phenotypic consequences of loss of homochirality in Pimt mutant flies. These mutants display reduced life span, contain protein aggregates, form melanotic tumors and display an enhanced susceptibility to tumor formation. Overall, this is a very interesting manuscript. The data are convincing and statistically verified. All the experiments are well explained and often graphically illustrated. The paper is also well written. I enjoyed reviewing this manuscript (and I don't say this often). Everybody knows from their biochemistry classes, that amino acids are incorporated into proteins in their L form. But at least I never thought about the consequences if the D form is used. The authors here generated genetic (the Pimt mutant) and diagnostic (the antibodies) tools to address this question. They specifically looked at caspases and their inability to cleave heterochiral substrates, but it is more than likely that many other biological processes are affected and give cause to disease. The antibodies generated by the authors will be of significant value for diagnostic purposes. There is a lot to be done, and the authors opened the door to it. I really only have a few formal comments/questions to further improve this manuscript:

We thank the Reviewer for the positive comments and suggestions.

1. In line 75, the authors report that the anti-DEVDG antibody had no reactivity in homochiral animals. They then explain, that the homochiral epitopes are not exposed at the surface in target proteins (line 77). How then do caspases recognise, bind and cleave their substrates under normal conditions? Isn't it also possible that this antibody does not work in immunofluorescent assays?

Previous studies have shown that certain caspase target sequences are only detectable immunologically following caspase-mediated proteolytic cleavage exposing neopeptides to the target protein surface (Tubulin Δ Csp6, cleaved caspase 3 or Fractin antibodies, Fan and Bergmann 2010, PMID: 19960024; Jennifer D Sokolowski 2014, PMID: 24507707), supporting the idea that our anti-DEVDG antibody is blind to

uncleaved sites. We agree with the Reviewer that it is important to provide background on this point and have added a literature review (please see page 5, lines 107-110).

2. They authors show that a significant fraction of the *Pimt* mutants have melanotic tumors. Certain Toll and Jak/Stat pathway mutants also form melanotic tumors. Could the observation that *Pimt* mutants form these tumors mean that loss of *Pimt* also deregulates Toll and/or Jak/Stat signaling?

We thank the Reviewer for this very interesting point, which we think is a likely scenario for the melanisation of the formed tumors in *Pimt* mutant animals. According to earlier work, melanotic tumors can result from either the overproliferation of blood cells or from an immune response toward abnormal cells and tissues in the larva. Indeed, formation of melanotic tumour masses was observed in various mutants affecting the Toll and JAK/STAT pathways and is one of the main phenotypic characteristics of *dcp-1* mutant animals. This was in fact one of the reasons we associated this phenotype to apoptosis and gave us the idea of using the expression of *tdcp1* for the assay (Song et al. 1997, PMID: 8999799).

We suggest that this phenotype is possibly resulting from a defect in cell death, similarly to *dcp-1* mutants which show consistent gut melanisation. In *dcp-1* mutants, no evidence for hyperplasia of the lymph glands or overproliferation of blood cells was found. Thus, the melanisation in these organs can be induced without the activation of the cellular immune response, or, more precisely, without the encapsulation response. In contrast, constitutive activation of the Jak/STAT pathway or overactivation of the Nf-kB/Rel pathway in the gain-of-function *Toll10B* mutant results in hemocyte proliferation, lamellocyte differentiation, and constitutive immune response (Luo et al. 2002, PMID: 11919715; Asha et al. 2003, PMID: 12586708; Agaisse and Perrimon 2004, PMID: 15199955; Zettervall et al. 2004, PMID: 15381778 and Qiu et al. 1998, PMID: 9550723 and reviewed by SMinakhina and Steward, 2006 PMID: 16816412). Addressing a potential coupling between caspases, immune and Jak/Stat pathways in the *Pimt* mutant phenotypes represents a full project on its own that we wish to explore in future work.

3. In lines 134/135, the authors write that “*Pimt*-RNAi cells had lowered caspase activity”. However, I think that is a misinterpretation. The caspase activity is likely unchanged, the heterochiral CD8-PARP-Venus reporter is just not cleaved anymore because of the structural change.

We fully agree with the Reviewer, we indeed mistakenly wrote caspase activity instead of “caspase activity was insufficient to cleave epimerised DEVDG substrates” (please see page 8, lines 182-183).

4. Introduction and Discussion are quite short. It would be nice if the authors can extend these a little.

We thank the Reviewer for this comment and completely agree that in the first version of the manuscript, Introduction and Discussion were incomplete. In the revised manuscript, we extended these sections (please see page 2, lines 35-38; pages 3-4, lines 48-68 and page 10, lines 218-221; pages 10-11; lines 226-245, respectively; see also response to Reviewer 1 and 3) and placed our new results in context with what is already known about *Pimt* in other organisms. We also emphasised and highlighted more the novelty of our results.

Reviewer: 3

D-amino acids have aroused interest in the past years since they have been detected in age-related processes, in particular D-aspartic acid residues in eye lens and brain. To study the effect of D-aspartyl residues the authors here present a plethora of experiments. In an in vitro assay they studied the effect of exchanging a critical Asp residue in a caspase-3 cleavage assay. MD calculations undermined that the D-variant is structurally hindered to enter the catalytic pocket of caspase-3. The authors also established a Drosophila model to study the effect of altered protein heterochiral repair by creating a Pimt null mutant. Pimt recognizes and converts D-aspartyl and iso-L aspartyl residues to the native L-aspartyl form within proteins. Pimt knock out flies died prematurely, but normal life span could be recovered by a Pimt wt knockin. Furthermore, they observed that progenitor cells (PCs) of the thermosensitive Notch-induced tumour model system NotchTS-RNAi flies developed tumours when Pimt was knocked out. The experiments are well-conceived and cover a wide range of methods.

Introduction and Discussion, however, give a less diligent impression. Both are very brief and contain commonplace arguments, but do not cite specific groundwork from the literature. Especially what is known on the effect of D-aspartic acid in proteins and how they are incorporated needs to be covered more extensively in the introduction. I suggest that in order to write a more thorough discussion, the authors report on established literature, e.g. what is known on reduction of lifespan by D-amino acids, and discuss what is new and exciting in their work compared to previous results and how their experiments expand previous knowledge.

We thank the Reviewer for the overall positive comments and apologise for the brevity of the introduction. In the revision, we extended these sections (please see page 2, lines 35-38; pages 3-4, lines 48-68 and page 10, lines 218-221; pages 10-11; lines 226-245) and placed our new results in context with what is already known about Pimt-mediated protein repair in other organisms. We also emphasised more the novelty of our results, as suggested.

A thorough revision of abbreviations and namings would also help. For example, Pimtn1#1/ Pimtn2#1 is not explained, neither is Pimtn1 #1/w1118. Labeling of antibodies is not always consistent in main text, Figure and Figure legend. E.g. the iso-L-form is denoted as DEVisoDG (text), anti-DEVisoDG (legend) and DEVbetaD (Figure). This is confusing for people not familiar with this sort of experiments.

Thank you for this important point, we are sorry about the confusion in our previous nomenclature. We have thoroughly revised abbreviations and naming in the new version of the manuscript (in all figures, legends and in the main text), and added a detailed list of genotypes (please see pages 40-41) and abbreviations (see page 66).

I must admit that I was a bit disappointed after reading title and abstract to find out that the authors only examine the effect of D-aspartic acid and not heterochirality or loss of homochirality in more general terms. Thus, I strongly advise to use a more moderate title and also adjust the abstract accordingly. The sentence "To address this question, we genetically altered protein homochirality in Drosophila and developed chiral-selective assays to detect protein-bound D-amino acids and assess their functional

significance.” is misleading in my eyes because you did not test any other amino acid apart from D-aspartic acid.

Among all chiral AA protein residues, l-aspartyl is the most unstable residue in proteins and it has the fastest isomerisation and epimerisation rate especially when present upstream of a glycine (Gly, G) residue. (Geiger & Clarke 1987, PMID: 3805008; Stephenson and Clarke 1989, PMID: 2703484; Capasso et al. 1991, PMID: 1823603; Bischoff et al. 1993, PMID: 8422378; Oliyai et al. 1994, PMID: 8058648). This explains the general predominance of isomerised and epimerised aspartyl residues among all non-l- α -amino acyl residues. We have now pointed this out more clearly in our manuscript and specified that our primary focus was on the effect of chirality changes affecting particularly aspartate (please see abstract on page 2, lines 22-23; introduction on pages 3-4, lines 48-68; results on page 4, lines 80-85; and discussion on page 10, lines 218-221).

REVIEWERS' COMMENTS

Reviewer #1 (Remarks to the Author):

The authors have addressed my comments adequately.

Reviewer #2 (Remarks to the Author):

The authors have addressed all my comments to my satisfaction. The authors spent extra effort to present their work clearer and have improved the manuscript. This is very interesting work and I have no further comments.

Reviewer #3 (Remarks to the Author):

The authors have thoroughly revised introduction and discussion. The manuscript's aim is much clearer now and also now summarizes previous literature. I have one problem with the manuscript at its current status. Upon request by reviewer 1, they performed an important control experiment that is now shown in Fig. 1b. They measured Pimt enzymatic activity on the heterochiral synthetic substrate. Here the control that shows that Pimt has no activity on the homochiral substrate must be shown as well. If space does not allow to add it to Figure 1, the control can easily be shown in the supplementary data. The experimental details about this experiment do not mention how errors were determined. This is likewise an important issue and must be added, especially as the errors scale almost linearly with time and only the upper error is given. Why does the curve show this peculiarity?

Response to reviewers – II.

We thank again all the three reviewers for their time in examining our manuscript and for providing their valuable comments. We have clarified the point raised by Reviewer #3 and discussed in detail the corresponding changes made to the final revised manuscript which are highlighted in yellow.

Reviewer #1 (Remarks to the Author):

The authors have addressed my comments adequately.

We thank the Reviewer for the valuable comments and suggestions which were very helpful and significantly improved the manuscript.

Reviewer #2 (Remarks to the Author):

The authors have addressed all my comments to my satisfaction. The authors spent extra effort to present their work clearer and have improved the manuscript. This is very interesting work and I have no further comments.

We thank the Reviewer for the valuable and generally very positive comments which helped to improve the manuscript.

Reviewer #3 (Remarks to the Author):

The authors have thoroughly revised introduction and discussion. The manuscript's aim is much clearer now and also now summarizes previous literature.

We thank the Reviewer for the useful and generally positive comments which helped to format the manuscript and make it more accessible to readers.

I have one problem with the manuscript at its current status.

Upon request by reviewer 1, they performed an important control experiment that is now shown in Fig. 1b. They measured Pimt enzymatic activity on the heterochiral synthetic substrate. Here the control that shows that Pimt has no activity on the homochiral substrate must be shown as well. If space does not allow to add it to Figure 1, the control can easily be shown in the supplementary data. The experimental details about this experiment do not mention how errors were determined. This is likewise an important issue and must be added, especially as the errors scale almost linearly with time and only the upper error is given. Why does the curve show this peculiarity?

We thank the Reviewer for pointing this important issue out. We have added the control experiment to the manuscript, which shows that Pimt has indeed no activity on the homochiral DEVDG substrate. We changed the way of data representation on Figure 1d., each data point as well as lower error bars are shown. We also pointed out in the figure legends the error bars and how they were determined. The peculiarity of the curve is expected to be linear as we used for our assays large excess of substrates to

ensure that substrate accessibility is not a limiting factor, thus product formation by the Pimt enzyme is linear with time.